# Exploring Large Action Sets with Hyperspherical Embeddings using von Mises-Fisher Sampling

**Walid Bendada** [1 2]  **Guillaume Salha-Galvan** [3]  **Romain Hennequin** [1]
**Théo Bontempelli** [1]  **Thomas Bouabça** [1]  **Tristan Cazenave** [2]

## Abstract

This paper introduces von Mises-Fisher exploration (vMF-exp), a scalable method for exploring large action sets in reinforcement learning problems where hyperspherical embedding vectors represent these actions. vMF-exp involves initially sampling a state embedding representation using a von Mises-Fisher distribution, then exploring this representation's nearest neighbors, which scales to virtually unlimited numbers of candidate actions. We show that, under theoretical assumptions, vMF-exp asymptotically maintains the same probability of exploring each action as Boltzmann Exploration (B-exp), a popular alternative that, nonetheless, suffers from scalability issues as it requires computing softmax values for each action. Consequently, vMF-exp serves as a scalable alternative to B-exp for exploring large action sets with hyperspherical embeddings. Experiments on simulated data, real-world public data, and the successful large-scale deployment of vMF-exp on the recommender system of a global music streaming service empirically validate the key properties of the proposed method.

## 1. Introduction

Exploration is a fundamental component of the reinforcement learning (RL) paradigm (Amin et al., 2021; McFarlane, 2018; Sutton & Barto, 2018). It allows RL agents to gather valuable information about their environment and identify optimal actions that maximize rewards (Amin et al., 2021; Chiappa et al., 2023; Dulac-Arnold et al., 2015; Jin et al., 2020; Ladosz et al., 2022; McFarlane, 2018; Reynolds,

2002; Slivkins et al., 2019; Sutton & Barto, 2018; Tang et al., 2017). However, as the set of actions to explore grows larger, the exploration process becomes increasingly challenging. Indeed, large action sets can lead to higher computational costs, longer learning times, and the risk of inadequate exploration and suboptimal policy development (Amin et al., 2021; Chen et al., 2021; Dulac-Arnold et al., 2015; Lillicrap et al., 2016; Sutton & Barto, 2018; Tomasi et al., 2023).

As an illustration, consider a recommender system on a music streaming service like Apple Music or Spotify, curating playlists of songs "inspired by" an initial selection to help users discover music (Bendada et al., 2023a). In practice, these services often generate such playlists all at once, using efficient nearest neighbor search systems (Johnson et al., 2019; Li et al., 2019) to retrieve songs most similar to the initial one, in a song embedding vector space learned using collaborative filtering or content-based methods (Bendada et al., 2023a;b; Bontempelli et al., 2022; Jacobson et al., 2016; Schedl et al., 2018; Zamani et al., 2019). Alternatively, one could formalize this task as an RL problem (Tomasi et al., 2023), where the recommender system (i.e., the agent) would adaptively select the next song to recommend (i.e., the next action) based on user feedback on previously recommended songs (i.e., the rewards, such as likes or skips). Using an RL approach instead of generating the playlist at once would have the advantage of dynamically learning from user feedback to identify the best recommendations (Afsar et al., 2022; Tomasi et al., 2023). However, music streaming services offer access to large catalogs with several millions of songs (Bendada et al., 2020; Jacobson et al., 2016; Schedl et al., 2018). Therefore, the agent would need to consider millions of possible actions for exploration, increasing the complexity of this task.

In particular, Boltzmann Exploration (B-exp) (Cesa-Bianchi et al., 2017; Sutton & Barto, 2018), one of the prevalent exploration strategies to sample actions based on embedding similarities, would become practically intractable as it would require computing softmax values over millions of elements (see Section 2). Furthermore, in large action sets, many actions are often irrelevant; in our example, most songs would constitute poor recommendations (Tomasi

[1]Deezer Research, Paris, France. [2]LAMSADE, Université Paris Dauphine, PSL, Paris, France. [3]SPEIT, Shanghai Jiao Tong University, Shanghai, China. Correspondence to: Walid Bendada <bendadaw@gmail.com>.

*Proceedings of the 42nd International Conference on Machine Learning*, Vancouver, Canada. PMLR 267, 2025. Copyright 2025 by the author(s).

et al., 2023). Therefore, random exploration methods like $\varepsilon$-greedy (Dann et al., 2022; Sutton & Barto, 2018), although more efficient than B-exp, would also be unsuitable for production use. Since these methods ignore song similarities, each song, including inappropriate ones, would have an equal chance of being selected for exploration. This could result in negative user feedback and a poor perception of the service (Tomasi et al., 2023). Lastly, deterministic exploration strategies would also be ineffective. Systems serving millions of users often rely on batch RL (Lange et al., 2012) since updating models after every trajectory is impractical. Batch RL, unlike on-policy learning, requires exploring actions non-deterministically given a state, and deterministic exploration would result in redundant trajectories and slow convergence (Bendada et al., 2020).

In summary, exploration remains challenging in RL problems characterized by large action sets and where accounting for embedding similarities is crucial, like our recommendation example. Overall, although a growing body of scientific research has been dedicated to adapting RL models for recommendation (see, e.g., the survey by Afsar et al. (2022)), evidence of RL adoption in commercial recommender systems exists but remains limited (Chen et al., 2019; 2021; 2022; Tomasi et al., 2023). The few existing solutions typically settle for a workaround by using a truncated version of B-exp (TB-exp). In TB-exp, a small subset of candidate actions is first selected, e.g., using approximate nearest neighbor search (a framework sometimes referred to as the Wolpertinger architecture (Dulac-Arnold et al., 2015)). Softmax values are then computed among those candidates only (Chen et al., 2019; 2021; 2022). YouTube, for instance, employs this technique for video recommendation (Chen et al., 2019). TB-exp allows for exploration in the close embedding neighborhood of a given state; however, it restricts the number of candidate actions based on technical considerations rather than optimal convergence properties. Although exploring beyond this restricted neighborhood might be beneficial, finding the best way to do so in large-scale settings remains an open research question.

In this paper[1], we propose to address this important question with a focus on its theoretical foundations. Our work examines the specific setting where actions are represented by embedding vectors with unit Euclidean norm, i.e., vectors lying on a unit hypersphere. As detailed in Section 2, this setting aligns with many real-world applications. Specifically, our contributions in this paper are as follows:

- We present von Mises-Fisher exploration (vMF-exp), a scalable sampling method for exploring large sets

of actions represented by hyperspherical embedding vectors. vMF-exp involves initially sampling a state embedding vector on the unit hypersphere using a von Mises-Fisher distribution (Fisher, 1953), then exploring the approximate nearest neighbors of this representation. This strategy effectively scales to large sets with millions of candidate actions to explore.

- We provide a comprehensive mathematical analysis of the behavior of vMF-exp, demonstrating that it exhibits desirable properties for effective exploration in large-scale RL problems. Notably, vMF-exp does not restrict exploration to a specific neighborhood and effectively preserves order while leveraging information from embedding vectors to assess action relevance.

- We also show that, under some theoretical assumptions, vMF-exp asymptotically maintains the same probability of exploring each action as the popular B-exp method while overcoming its scalability issues. This positions vMF-exp as a scalable alternative to B-exp.

- The primary objective of this paper is the introduction and mathematical analysis of the theoretical properties of vMF-exp. Nonetheless, as a complement to this analysis, we also report empirical validations of the method, including experiments on simulated data, on real-world publicly available data, and a discussion of the recent and successful deployment of vMF-exp on a global music streaming service to recommend music to millions of users daily. These experiments validate the key properties of the proposed method and its potential.

- We publicly release a Python implementation of vMF-exp on GitHub to enable reproducibility of our experiments and to encourage future use of the method: https://github.com/deezer/vMF-exploration.

The remainder of this paper is organized as follows. Section 2 formalizes the problem. Section 3 introduces vMF-exp, Section 4 details our theoretical analysis, Section 5 discusses our experiments, and Section 6 concludes.

## 2. Preliminaries

We begin this section by formally introducing the problem addressed in this paper, followed by an explanation of the limitations of existing and popular RL exploration strategies.

### 2.1. Problem Formulation

**Notation** Throughout this paper, we consider an RL agent sequentially selecting actions within a set of $n \in \mathbb{N}^{\star}$ actions:

$$\mathcal{I}_n = \{1, 2, \ldots, n\}. \tag{1}$$

---

[1]Parts of the content published in this ICML 2025 conference paper were previously presented at two non-archival workshops: ICML 2024 ARLET (Bendada et al., 2024) and ICLR 2025 FPI (Bendada et al., 2025).

Each action $i \in \mathcal{I}_n$ is represented by a distinct low-dimensional vectorial representation $X_i \in \mathbb{R}^d$, i.e., by an embedding vector or simply an embedding[2], for some fixed dimension $d \in \mathbb{N}$ with $d \geq 2$ and $d \ll n$. Additionally, we assume all vectors have a unit Euclidean norm, i.e., $\|X_i\|_2 = 1, \forall i \in \mathcal{I}_n$. They form a set of embeddings noted $\mathcal{X}_n = \{X_i\}_{1 \leq i \leq n} \in (\mathcal{S}^{d-1})^n$, where $\mathcal{S}^{d-1}$ is the $d$-dimensional unit hypersphere (Fisher, 1953):

$$\mathcal{S}^{d-1} = \left\{ x \in \mathbb{R}^d : \|x\|_2 = 1 \right\}. \tag{2}$$

We also assume the availability of an approximate nearest neighbor (ANN) (Johnson et al., 2019; Li et al., 2019) search engine. Using this engine, for any vector $V \in \mathcal{S}^{d-1}$, the nearest neighbor of $V$ among $\mathcal{X}_n$ in terms of inner product similarity (equal to the cosine similarity, for unit vectors (Tan et al., 2016)), called $X_{i_V^\star}$, can be retrieved in a sublinear time complexity with respect to $n$. Although ANN engines are parameterized based on a trade-off between efficiency and accuracy, we make the simplifying assumption that $X_{i_V^\star}$ is the actual nearest neighbor of $V$, which we later discuss in Section 4.3. Formally:

$$i_V^\star = \arg\max_{i \in \mathcal{I}_n} \langle V, X_i \rangle. \tag{3}$$

Returning to the illustrative example of Section 1, $\mathcal{X}_n$ would represent embeddings associated with each song of the catalog $\mathcal{I}_n$ of the music streaming service. In this case, $n$ would be on the order of several millions (Bendada et al., 2020; Briand et al., 2021; Jacobson et al., 2016). The RL agent would be the recommender system sequentially recommending these songs to users. Normalizing embeddings is a common practice in both academic and industrial recommender systems (Afchar et al., 2023; Bontempelli et al., 2022; Kim et al., 2023; Schedl et al., 2018) to mitigate popularity biases, as vector norms often encode popularity information on items (Afchar et al., 2023; Chen et al., 2023). Normalizing embeddings also prevents inner products from being unbounded, avoiding overflow and underflow numerical instabilities (LeCun et al., 2015).

At time $t$, the agent considers a state vector $V_t \in \mathcal{S}^{d-1}$, noted $V$ for brevity. It selects the next action in $\mathcal{I}_n$, whose relevance is evaluated by a reward provided by the environment. In our example, the agent would recommend the next song to continue the playlist, based on the previous song whose embedding $V$ acts as the current state. In this case, the reward might be based on user feedback, such as liking or skipping the song (Bontempelli et al., 2022). The agent may select $i_V^\star$, i.e., exploit $i_V^\star$ (Sutton & Barto, 2018).

___
[2]At this stage, we do not make assumptions regarding the specific methods used to learn these embedding vectors, nor the interpretation of proximity between vectors in the embedding space.

Alternatively, it may rely on an exploration strategy to select another $\mathcal{I}_n$ element. Formally, an exploration strategy $P$ is a policy function (Sutton & Barto, 2018) that, given $V$, selects each action $i \in \mathcal{I}_n$ with a probability $P(i \mid V) \in [0, 1]$.

**Objective**  Our goal in this paper is to develop a suitable exploration strategy for our specific setting, where hyperspherical embedding vectors represent actions, and the number of actions can reach millions. Precisely, we aim to obtain an exploration scheme meeting the following properties:

- Scalability (**P1**): we consider an exploration scalable if the time required to sample actions given a vector $V$ is at most the time needed for the ANN engine to retrieve the nearest neighbor, which is typically achieved in a sublinear time complexity with respect to $n$. Scalability is a mandatory requirement for exploring large action sets with millions of elements.

- Unrestricted radius (**P2**): Radius$(P \mid V)$ is the number of actions with a non-zero probability of being explored given a state $V$. While exploring actions too far from $V$ might be suboptimal (e.g., resulting in poor recommendations), it is crucial that exploration is not restricted to a specific radius by construction. Such a restriction could prevent the agent from exploring relevant actions that lie beyond this radius. An unrestricted radius ensures that the exploration strategy remains flexible and capable of adapting to various contexts, allowing for the exploration of relevant actions regardless of their embedding position.

- Order preservation (**P3**): order is preserved when the probability of selecting the action $i$ given the state $V$ is a strictly increasing function of $\langle V, X_i \rangle$. More formally, order preservation requires that, $\forall (i, j) \in \mathcal{I}_n^2$,

$$\langle V, X_i \rangle > \langle V, X_j \rangle \implies P(i \mid V) > P(j \mid V). \tag{4}$$

  **P3** ensures that the exploration strategy properly leverages the information captured in the embedding vectors to assess the relevance of an action given a state.

### 2.2. Limitations of Existing Exploration Strategies

Finding an strategy that simultaneously meets the three properties **P1**, **P2** and **P3** is essential for effective exploration in RL problems with large action sets and embedding representations. Nonetheless, existing exploration methods fail to achieve this, which motivates our work in this paper.

**Random and $\varepsilon$-greedy Exploration**  The most straightforward example of an action exploration strategy would be the random (uniform) policy, where:

$$\forall i \in \mathcal{I}_n, P_{\text{rand}}(i \mid V) = \frac{1}{n}. \tag{5}$$

A popular variant is the $\varepsilon$-greedy strategy (Sutton & Barto, 2018). With a probability $\varepsilon \in [0, 1]$, the agent would choose the next action uniformly at random. With a probability $1 - \varepsilon$, it would exploit the most relevant action based on its knowledge. Random and $\varepsilon$-greedy exploration strategies are scalable (**P1**), as elements of $\mathcal{I}_n$ can be uniformly sampled in $\mathcal{O}(1)$ time (Cormen et al., 2022). Additionally, they verify **P2**. Indeed, Radius$(P_{\text{rand}}|V) = n$ since every action can be selected. However, these strategies ignore embeddings at the sampling phase and do not achieve order preservation (**P3**). This is a significant limitation, reinforced by the fact that these policies have a maximal radius. As explained in Section 1, in large action sets, many actions are often irrelevant, e.g., most songs from the musical catalog would constitute poor recommendations given an initial state (Tomasi et al., 2023). Exploring each action/song with equal probability, including inappropriate ones, could result in negative user feedback and a poor perception of the service (Tomasi et al., 2023).

**Boltzmann Exploration**  To overcome the limitations of random exploration, actions can be sampled based on their embedding similarity with $V$, typically measured using dot products. The prevalent approach in RL is Boltzmann Exploration (B-exp) (Amin et al., 2021; Cesa-Bianchi et al., 2017; Chen et al., 2021; Sutton & Barto, 2018), which employs the Boltzmann distribution for action sampling:

$$\forall i \in \mathcal{I}_n, P_{\text{B-exp}}(i \mid V, \mathcal{X}_n, \kappa) = \frac{e^{\kappa\langle V, X_i\rangle}}{\sum_{j=1}^{n} e^{\kappa\langle V, X_j\rangle}}, \quad (6)$$

where the hyperparameter $\kappa \in \mathbb{R}^+$ controls the entropy of the distribution. B-exp samples actions according to a strictly increasing function of their inner product similarity with $V$ for $\kappa > 0$, guaranteeing order preservation (**P3**). By carefully tuning $\kappa$, one can ensure that irrelevant actions are practically never selected while maintaining a non-zero probability of recommending actions with less than maximal similarity, thereby indirectly controlling the radius of the policy (**P2**). Unfortunately, B-exp does not satisfy **P1**, i.e., it is not scalable to large action sets. Indeed, evaluating Equation (6) requires explicitly computing the probability of sampling each individual action before actually sampling from them, which is prohibitively expensive for large values of $n$ (Chen et al., 2021). Note that, while we focus on B-exp, these concerns would remain valid for any other sampling distribution requiring explicitly computing similarities and probabilities for each of the $n$ actions (Amin et al., 2021).

**Truncated Boltzmann Exploration**  Due to these scalability concerns, previous work sometimes settled for a workaround consisting in sampling actions from a Truncated version of B-exp (Chen et al., 2021), which we refer to as TB-exp. A small number $m \ll n$ of candidate actions,

usually around hundreds or thousands, is first retrieved using the ANN search engine, leading to a candidate action set $\mathcal{I}_m(V)$. The sampling step is subsequently performed only within $\mathcal{I}_m(V)$. More formally, for all $i \in \mathcal{I}_m(V)$:

$$P_{\text{TB-exp},m}(i \mid V, \mathcal{X}_n, \kappa) = \frac{e^{\kappa\langle V, X_i\rangle}}{\sum_{j\in\mathcal{I}_m(V)} e^{\kappa\langle V, X_j\rangle}}. \quad (7)$$

TB-exp performs action selection in a time that depends on $m$ instead of $n$, and has been successfully deployed in production environments involving millions of actions (Chen et al., 2019; 2021; 2022). While it still satisfies **P3**, TB-exp also meets **P1** for small values of $m$. However, it no longer satisfies **P2**. This method restricts the radius, i.e., the number of candidate actions, based on technical considerations rather than exploration efficiency. This restriction can potentially hinder model convergence by neglecting the exploration of relevant actions beyond this fixed radius. This highlights the difficulty of designing a method that simultaneously satisfies **P1**, **P2**, and **P3** – ideally, one with properties akin to B-exp but with greater scalability.

## 3. From Boltzmann to vMF Exploration

In this section, we present our proposed solution for exploring large action sets with hyperspherical embeddings.

### 3.1. von Mises–Fisher Exploration

The inability of B-exp to scale arises from its need to compute all $n$ sampling probabilities explicitly. In this paper, we propose von Mises-Fisher Exploration (vMF-exp), an alternative strategy that overcomes this constraint. Specifically, given an initial state vector $V$, vMF-exp consists in:

- Firstly, sampling an hyperspherical vector $\tilde{V}$ on the unit hypersphere $\mathcal{S}^{d-1}$, according to a von Mises-Fisher distribution (Fisher, 1953) centered on $V$.

- Secondly, selecting $\tilde{V}$'s nearest neighbor action in the embedding space for exploration.

In directional statistics, the vMF distribution (Fisher, 1953) is a continuous vector probability distribution defined on the unit hypersphere $\mathcal{S}^{d-1}$. It has recently been used in RL to assess the uncertainty of gradient directions (Zhu et al., 2024). For all $\tilde{V} \in \mathcal{S}^{d-1}$, its probability density function (PDF) is defined as follows:

$$f_{\text{vMF}}(\tilde{V} \mid \kappa, V, d) = C_d(\kappa)e^{\kappa\langle V, \tilde{V}\rangle}, \quad (8)$$

with:

$$C_d(\kappa) = \frac{1}{\int_{\tilde{V}\in\mathcal{S}^{d-1}} e^{\kappa\langle V, \tilde{V}\rangle}\, \mathrm{d}\tilde{V}} = \frac{\kappa^{\frac{d}{2}-1}}{(2\pi)^{\frac{d}{2}} I_{\frac{d}{2}-1}(\kappa)}. \quad (9)$$

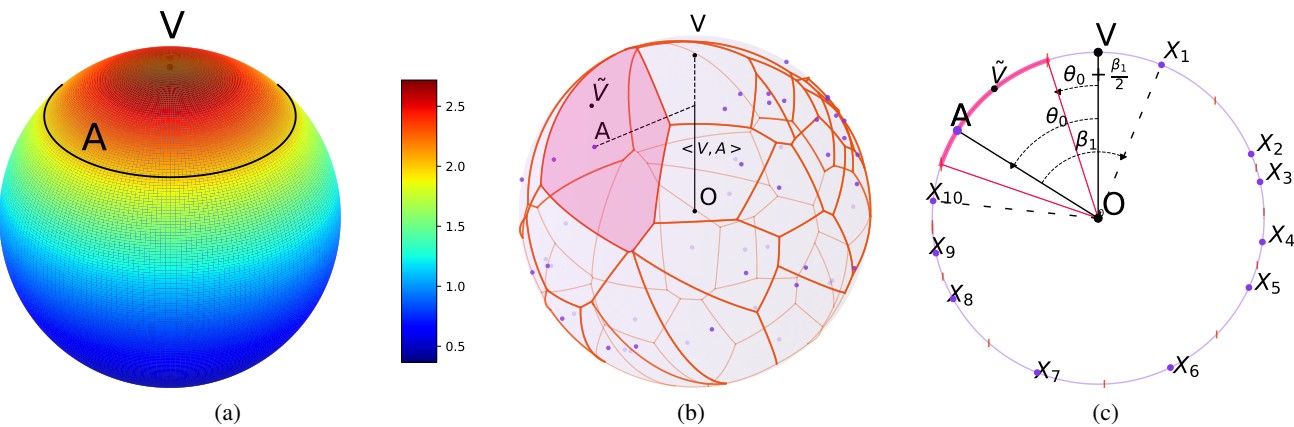

(a)                                (b)                                (c)

*Figure 1.* (a) Probability density function (PDF) of a 3-dimensional vMF distribution. (b) vMF-exp explores the action $a$ represented by the embedding vector $A$ when the sampled vector $\tilde{V}$ lies in $A$'s Voronoï cell, shown in red in 3D. (c) Same as (b) in a 2-dimensional setting.

and where $\kappa \in \mathbb{R}^+$. The function $I_{\frac{d}{2}-1}$ designates the modified Bessel function of the first kind (Baricz, 2010) at order $d/2 - 1$. Figure 1(a) illustrates the PDF of a vMF distribution on the 3-dimensional unit sphere. For any $\tilde{V} \in \mathcal{S}^{d-1}$, $f_{\text{vMF}}(\tilde{V} \mid \kappa, V, d)$ is proportional to $e^{\kappa\langle V, \tilde{V}\rangle}$, which is reminiscent of the B-exp sampling probability of Equation (6). The hyperparameter $\kappa$ controls the entropy of the distribution. In particular, for $\kappa = 0$, the vMF distribution boils down to the uniform distribution on $\mathcal{S}^{d-1}$.

### 3.2. Properties

**P1** vMF-exp only requires sampling a $d$-dimensional vector instead of handling a discrete distribution with $n$ parameters, allowing $\tilde{V}$ to be sampled in constant time with respect to $n$. Therefore, vMF-exp is a scalable exploration strategy. Efficient sampling algorithms for vMF distributions have been well-studied (Kang & Oh, 2024; Pinzón & Jung, 2023) (see Appendix F for practical details). As shown in the following sections, we successfully explored sets of millions of actions without scalability issues, using the Python vMF sampler from Pinzón & Jung (2023) for simulations in Section 4 and a custom implementation for the recommendation application in Section 5.

**P2** The probability of sampling $i \in \mathcal{I}_n$ given $V$ for exploration is the probability that $X_i$ is the nearest neighbor of $\tilde{V}$ within $\mathcal{X}_n$, i.e., that $\tilde{V}$ lies in $\mathcal{S}_{\text{Voronoï}}(X_i \mid \mathcal{X}_n)$, the Voronoï cell of $X_i$ in the Voronoï tessellation of $\mathcal{S}^{d-1}$ defined by $\mathcal{X}_n$ (Du et al., 1999; 2010) (see Figures 1(b), 1(c)). We have:

$$\mathcal{S}_{\text{Voronoï}}(X_i \mid \mathcal{X}_n) = \{\tilde{V} \in \mathcal{S}^{d-1}, \forall j \in \mathcal{I}_n, \\ \langle \tilde{V}, X_i \rangle \geq \langle \tilde{V}, X_j \rangle\}, \quad (10)$$

and:

$$\bigcup_{i \in \mathcal{I}_n} \mathcal{S}_{\text{Voronoï}}(X_i \mid \mathcal{X}_n) = \mathcal{S}^{d-1}. \quad (11)$$

Thus, vMF-exp's sampling probabilities can be written as:

$$\forall i \in \mathcal{I}_n, P_{\text{vMF-exp}}(i \mid V, \mathcal{X}_n, \kappa) = \\ \int_{\tilde{V} \in \mathcal{S}_{\text{Voronoï}}(X_i \mid \mathcal{X}_n)} f_{\text{vMF}}(\tilde{V} \mid \kappa, V, d)\, \mathrm{d}\tilde{V}, \quad (12)$$

which is always strictly positive. Therefore, vMF-exp satisfies the unrestricted radius property (**P2**). Similar to B-exp, adjusting $\kappa$ ensures that actions with low similarity have negligible sampling probabilities in practice.

**P3** $P_{\text{vMF-exp}}(i \mid V, \mathcal{X}n, \kappa)$ increases due to two factors: the average $f_{\text{vMF}}(\tilde{V} \mid \kappa, V, d)$ value for $\tilde{V} \in \mathcal{S}_{\text{Voronoï}}(X_i \mid \mathcal{X}_n)$, correlated to $\langle X_i, V \rangle$ and contributing to order preservation, and the surface area of $\mathcal{S}_{\text{Voronoï}}(X_i \mid \mathcal{X}_n)$, which measures $X_i$'s dissimilarity to other $\mathcal{X}_n$ elements. Actions in a low-density subspace of $\mathcal{S}^{d-1}$ have larger Voronoï cells and may be selected more often than those near $V$ but in high-density regions. Thus, vMF-exp favors actions similar to $V$ and dissimilar to others, with order preservation being dependent on $\mathcal{X}_n$'s distribution. Section 4 focuses on a setting where B-exp and vMF-exp asymptotically share similar probabilities. Consequently, vMF-exp, like B-exp, will verify order preservation (**P3**). In conclusion, in this setting, vMF-exp will verify **P1**, **P2**, and **P3** simultaneously.

## 4. Theoretical Comparison: vMF-exp vs B-exp

We now provide a mathematical comparison of vMF-exp and B-exp. We focus on the theoretical setting presented in Section 4.1. We show that, in this setting, vMF-exp maintains the same probability of exploring each action as B-exp, while overcoming its scalability issues. As noted above, this implies that vMF-exp verifies **P1**, **P2**, and **P3** simultaneously and, therefore, acts as a scalable alternative to the popular but unscalable B-exp for exploring large action sets with hyperspherical embeddings.

## 4.1. Setting and Assumptions

We focus on the setting where embeddings are independent and identically distributed (i.i.d.) and follow a uniform distribution on the unit hypersphere, i.e.,

$$\mathcal{X}_n \sim \mathcal{U}(\mathcal{S}^{d-1}). \tag{13}$$

For convenience in our proofs, we consider the action set to be the union of $\mathcal{I}_n$, the set of $n$ actions, and another action $a$ with a known embedding $A \in \mathcal{S}^{d-1}$. The resulting entire action set $\mathcal{I}_{n+1}$ and embedding set $\mathcal{X}_{n+1}$ are defined as $\mathcal{I}_{n+1} = \mathcal{I}_n \cup \{a\}$ and $\mathcal{X}_{n+1} = \mathcal{X}_n \cup \{A\}$. In this section, we are interested in the probability of each exploration scheme, B-exp and vMF-exp, to sample $a$ among all actions of $\mathcal{I}_{n+1}$ given a state embedding vector $V \in \mathcal{S}^{d-1}$. These probabilities are defined respectively as:

$$P_{\text{B-exp}}(a \mid n, d, V, \kappa) =$$
$$\mathbb{E}_{\mathcal{X}_n \sim \mathcal{U}(\mathcal{S}^{d-1})} \Big[ P_{\text{B-exp}}(a \mid V, \mathcal{X}_{n+1}, \kappa) \Big], \tag{14}$$

and:

$$P_{\text{vMF-exp}}(a \mid n, d, V, \kappa) =$$
$$\mathbb{E}_{\mathcal{X}_n \sim \mathcal{U}(\mathcal{S}^{d-1})} \Big[ P_{\text{vMF-exp}}(a \mid V, \mathcal{X}_{n+1}, \kappa) \Big]. \tag{15}$$

## 4.2. Results

We now present and discuss our main theoretical results. For brevity, we report all intermediary lemmas and mathematical proofs in the Appendices A, B, C and D of this paper. Our first and most general result links the asymptotic behavior of B-exp and vMF-exp as the action set grows.

**Proposition 4.1.** *In the setting of Section 4.1, we have:*

$$\lim_{n \to +\infty} \frac{P_{\text{B-exp}}(a \mid n, d, V, \kappa)}{P_{\text{vMF-exp}}(a \mid n, d, V, \kappa)} = 1. \tag{16}$$

Proposition 4.1 states that, for large values of $n$, the probability of sampling the action $a$ for exploration is asymptotically the same using either B-exp or vMF-exp. This result follows from the respective asymptotic characterizations of $P_{\text{B-exp}}$ and $P_{\text{vMF-exp}}$, detailed below. Importantly, it implies that, for large values of $n$, vMF-exp shares the same properties as B-exp (**P2**, **P3**), including order preservation. However, as noted in Section 3, vMF-exp offers greater scalability since its implementation only requires sampling a vector of a fixed size $d$, an operation independent of $n$ (**P1**).

Next, we present a common approximate analytical expression for both methods, denoted $P_0$ and defined as follows:

$$P_0(a \mid n, d, V, \kappa) = \frac{f_{\text{vMF}}(A \mid V, \kappa) \mathcal{A}(\mathcal{S}^{d-1})}{n}, \tag{17}$$

with $\mathcal{A}(\mathcal{S}^{d-1})$ denoting the surface area of the hypersphere $\mathcal{S}^{d-1}$. The following two propositions describe the rate at which this asymptotic behavior is reached by B-exp and vMF-exp, respectively, as $n$ grows.

**Proposition 4.2.** *In the setting of Section 4.1, we have:*

$$P_{\text{B-exp}}(a \mid n, d, V, \kappa) = P_0(a \mid n, d, V, \kappa)$$
$$+ o(\frac{1}{n\sqrt{n}}). \tag{18}$$

**Proposition 4.3.** *In the setting of Section 4.1, we have:*

$$P_{\text{vMF-exp}}(a \mid n, d, V, \kappa) = P_0(a \mid n, d, V, \kappa)$$
$$+ \begin{cases} \mathcal{O}(\frac{1}{n^2}) & \text{if } d = 2, \\ \mathcal{O}(\frac{1}{n^{1+\frac{2}{d-1}}}) & \text{if } d > 2. \end{cases} \tag{19}$$

In essence, when $n$ is large, the probability of sampling $a$ can be approximated by the PDF of the vMF distribution evaluated at $A$ multiplied by the average surface area of $A$'s Voronoï cell, for both methods. As $n$ grows, this Voronoï cell shrinks until $f_{\text{vMF}}$ becomes nearly constant across its entire surface. Figure 2(f) illustrates this interpretation.

The rate at which the two exploration methods reach their asymptotic behavior differs. Specifically, the shrinking rate of the Voronoi cell depends on the dimension of the hypersphere, explaining why the second term in Equation (19) depends on $d$. This dependency does not occur with B-exp. Consequently, for large values of $d$, one may require a higher number of actions $n$ before the asymptotic behavior of Equation (16) is observed. For this reason, it is useful to obtain a more precise approximation of $P_{\text{vMF-exp}}(a \mid n, d, V, \kappa)$ when $d$ increases, which we provide in the next section.

## 4.3. Discussion

**High Dimension** Building on the above discussion, Proposition 4.4 provides a more precise expression for $P_{\text{vMF-exp}}(a \mid n, V, \kappa)$ when $d$ increases (approximately $d \geq 20$ in our experiments). This expression is derived by examining the first two terms of the Taylor expansion (Abramowitz & Stegun, 1948) of $f_{\text{vMF}}$ near $A$, rather than relying solely on the zero-order term. The second term becomes increasingly significant as $d$ grows. Despite its apparent complexity, the expression has a straightforward interpretation: the negative sign before $\langle V, A \rangle$ indicates that, when $A$ is similar to $V$, it is sampled less often than with B-exp for the same $\kappa$ and $d$. Conversely, when $A$ is on the opposite side of the hypersphere, the term positively contributes to $P_{\text{vMF-exp}}(a \mid n, V, \kappa)$. In summary, for larger $d$, vMF-exp is expected to explore more extensively than B-exp with the same $\kappa$.

**Proposition 4.4.** *Let $B : (z_1, z_2) \mapsto \int_0^1 t^{z_1-1}(1-t)^{z_2-1} \, dt$ denote the Beta function, and $\Gamma : z \mapsto \int_0^\infty t^{z-1}e^{-t} \, dt$*

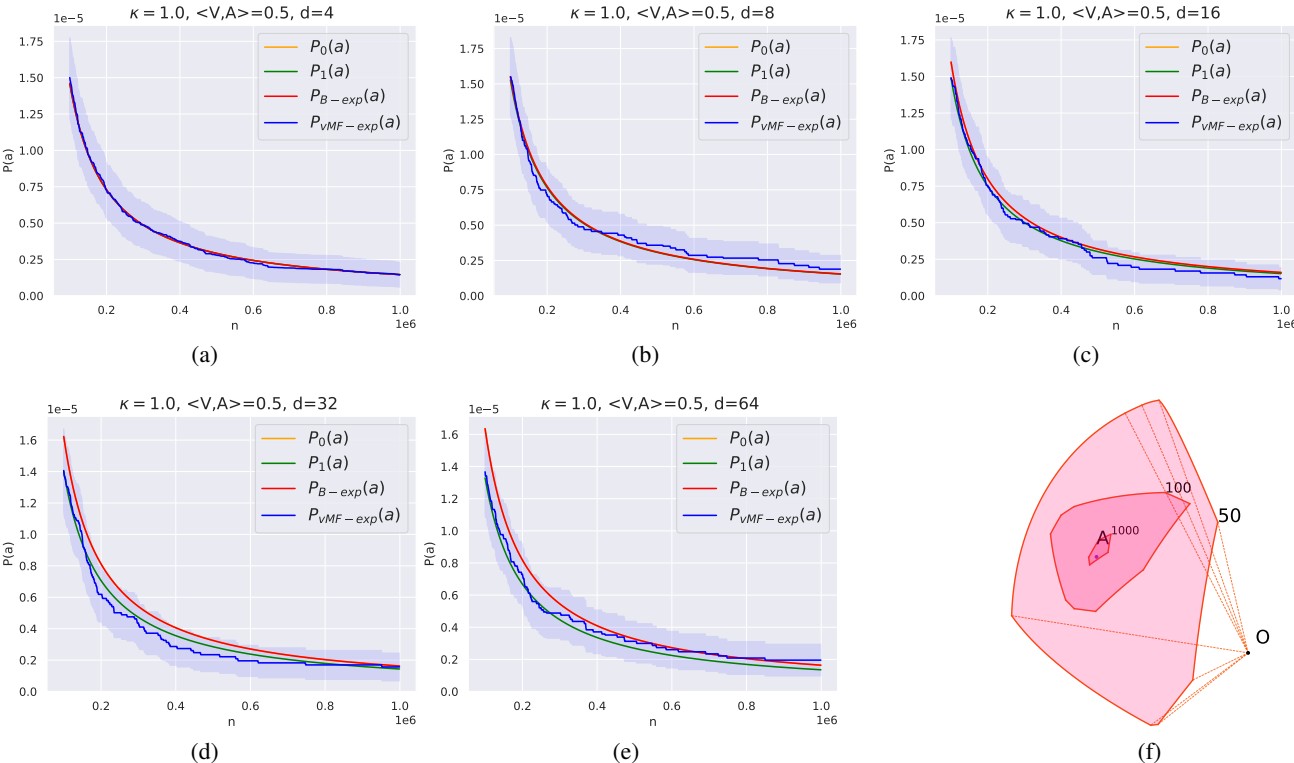

Figure 2. (a) to (e): Validation of the key properties discussed in Sections 4.2 and 4.3 using Monte Carlo simulations, as further elaborated in Section 4.3. (f) Illustration of the 3-dimensional Voronoï cell of a vector $A$, for action numbers $n \in \{50, 100, 1000\}$.

*denote the Gamma function (Abramowitz & Stegun, 1948). In the setting of Section 4.1 with $d \geq 3$, we have:*

$$P_{vMF\text{-}exp}(a \mid n, V, \kappa) = P_1(a \mid n, V, \kappa) \\ + \mathcal{O}(\frac{1}{n^{\frac{2}{d-1}}}), \quad (20)$$

*with:*

$$P_1(a \mid n, V, \kappa) = P_0(a \mid n, V, \kappa) \\ - \left[ \frac{f_{vMF}(A \mid V, \kappa) \mathcal{A}(\mathcal{S}^{d-1})}{n} \frac{\kappa \langle V, A \rangle \Gamma(\frac{d+1}{d-1})}{2} \\ \times \left( \frac{(d-1)B(\frac{1}{2}, \frac{d-1}{2})}{n} \right)^{\frac{2}{d-1}} \right]. \quad (21)$$

**The case $d = 2$** In 2 dimensions, Voronoï cells are arcs of a circle and are delimited by the perpendicular bisectors of two neighboring points, as shown in Figure 1(c). Interestingly, in this specific case, $P_{vMF\text{-}exp}(a \mid n, d = 2, V, \kappa)$ can be computed using geometric arguments. A comprehensive mathematical analysis is provided in Appendix B. This analysis confirms that, when $d = 2$, vMF-exp approaches its asymptotic behavior faster than B-exp, as indicated by the $\mathcal{O}(\frac{1}{n^2})$ term in Proposition 4.3.

**Validation of Theoretical Properties via Monte Carlo Simulations** Using the efficient Python sampler of Pinzón & Jung (2023), we repeatedly sampled vectors $\mathcal{X}_n \sim \mathcal{U}(S^{d-1})$ and $\tilde{V} \sim \text{vMF}(V, \kappa)$, for various values of $d, \kappa$, and $\langle V, A \rangle$. Figure 2 reports, for $\kappa = 1.0$, $\langle V, A \rangle = 0.5$ and growing values of $d$, the $P_{vMF\text{-}exp}(a)$ sampling probability depending on the number of actions $n$, as well as $P_{B\text{-}exp}(a)$ with similar parameters and our approximations $P_0(a)$ and $P_1(a)$. We repeated all experiments 8 million times and reported 95% confidence intervals. The results are consistent with our key theoretical findings in this paper.

Firstly, in line with Proposition 4.2, $P_{B\text{-}exp}(a)$ and $P_0(a)$ are indistinguishable for this range of $n$ values. Secondly, for small $d$ values (Figures 2(a), 2(b), 2(c)), $P_{vMF\text{-}exp}$ is also tightly aligned with $P_{B\text{-}exp}(a)$ and $P_0(a)$, consistently with Proposition 4.1 and 4.3. Note that the y-axis is on a 1e-5 scale; hence, probabilities are extremely close. Thirdly, when $d \geq 16$ (Figures 2(d), 2(e)), $P_1(a)$ becomes more distinguishable from $P_0(a)$ and constitutes a better approximation of $P_{vMF\text{-}exp}(a)$ than $P_0(a)$, as per Proposition 4.4. Lastly, since $\langle V, A \rangle > 0$, Proposition 4.4 predicts that $P_{B\text{-}exp}(a) \geq P_{vMF\text{-}exp}(a)$ for large $d$, which our experiments confirm. We provide comparable simulations with other $(d, \kappa, \langle V, A \rangle)$ combinations in Appendix G. All simulations are reproducible using our source code (see Section 5).

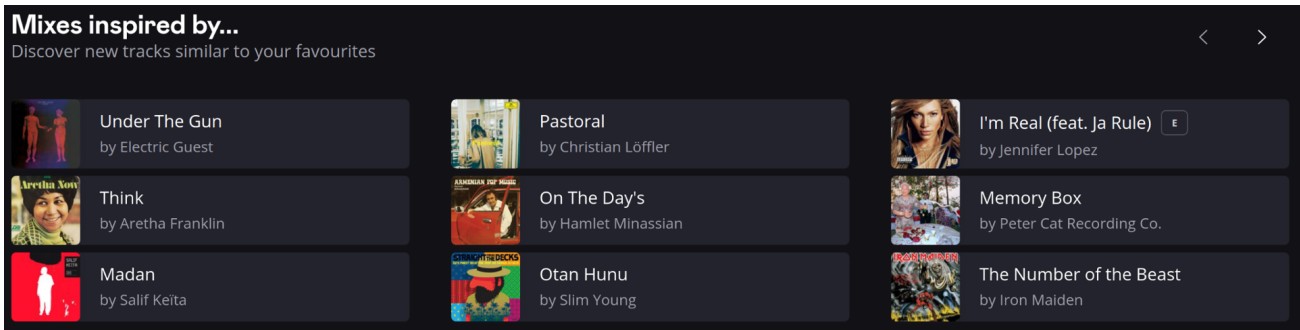

*Figure 3.* Interface of the "Mixes Inspired By" recommender system on the music streaming service Deezer. This system presents a personalized shortlist of songs liked by the user. Clicking on a song generates a playlist "inspired by" this song. As detailed in Section 5 and Appendix I, vMF-exp has been employed for months on the production environment of this service to generate playlists, exploring songs from a catalog containing millions of candidates.

**Link with Thompson Sampling**   One might notice interesting similarities between vMF-exp and bandit arm exploration with the Thompson Sampling method (Chapelle & Li, 2011). We refer the interested reader to our Appendix E for a detailed comparison of the two approaches.

**Limitations and Future Work**   While we believe our study provides valuable insights into vMF-exp, several limitations must be acknowledged. Our theoretical guarantees are currently restricted to the distributions described in Section 4.1. Although vMF-exp can be applied in practice with hyperspherical embeddings from any other distributions, we have not yet extended our guarantees to such cases[3].

For instance, studying vMF-exp in clustered embedding settings, as is sometimes the case with music recommendation embeddings (where clusters can, e.g., summarize music genres (Salha-Galvan et al., 2022)), could be insightful. We believe that future research should benefit from the approach we proposed in this paper to derive the full (non-trivial) demonstration for the uniform distribution case.

Our future work will also investigate the second-order term from Proposition 4.4, which may be significant for large $\kappa$, as well as the impact of ANN errors. Although our analysis assumes exact neighbor retrieval, this assumption may break down for extremely large action sets (Johnson et al., 2019), potentially causing minor exploration perturbations.

## 5. vMF-exp in the Real World

As explained in Section 1, the primary objective of this paper was to introduce vMF-exp and provide a rigorous mathematical analysis of its theoretical properties for exploring large action sets represented by hyperspherical embedding vectors. Nonetheless, as an opening to our work and a complement to this mathematical analysis, we describe in this section several empirical validations of vMF-exp, further detailed in Appendices G, H, and I. Taken together, these experiments validate the claimed scalability and theoretical properties of the proposed method, as well as its potential and impact for real-world applications. Specifically:

- To begin with, we recall that, as explained in Section 4.3, a more comprehensive outline of the Monte Carlo simulations discussed in that section is provided in Appendix G. The additional results presented in this appendix are consistent with those shown in Figure 2 of Section 4.3. To ensure the reproducibility of these experiments and to encourage future use of the vMF-exp method, we have released a Python implementation of vMF-exp on GitHub with this paper: `https://github.com/deezer/vMF-exploration`.

- To go further, we recognize that some readers may wish to explore our topic through reproducible experiments on real-world data. Therefore, in Appendix H, we empirically validate the main properties of vMF-exp using a large-scale, publicly available dataset of one million GloVe word embedding vectors (Pennington et al., 2014). We demonstrate that vMF-exp simultaneously satisfies **P1**, **P2**, and **P3** on this GloVe dataset. Furthermore, this study on GloVe vectors shows the accuracy of our approximations of $P_{\text{B-exp}}(a)$ and $P_{\text{vMF-exp}}(a)$ from Propositions 4.2, 4.3, and 4.4, despite the fact that GloVe vectors do not strictly meet the i.i.d. and uniform assumptions of our theoretical study. This additional study is fully reproducible using the code provided in the GitHub repository mentioned above.

- The final appendix of this paper, Appendix I, showcases a real-world application of vMF-exp. We present its successful large-scale deployment in the private

---

[3]Nonetheless, results from Section 5 will tend to confirm the practical usefulness of our propositions on real-world embedding vectors that do *not* strictly satisfy assumptions from Section 4.1.

production system of the global music streaming service Deezer (Bendada et al., 2023a). On this service, vMF-exp has been employed for months to recommend playlists of songs inspired by an initial selection to millions of users (see Figure 3), exploring a catalog of millions of candidate songs. This application, validated by a positive worldwide online A/B test, confirms the practical relevance of our work. As vMF-exp was successfully deployed in production, achieving a sampling latency of just a few milliseconds, it also confirms the scalability of the method.

## 6. Conclusion

This paper introduced vMF-exp, a scalable method for exploring large action sets in RL problems where hyperspherical embedding vectors represent these actions. vMF-exp scales effectively to large sets with millions of actions, exhibits desirable properties, and, under theoretical assumptions, even asymptotically preserves the same exploration probabilities as B-exp, a prevalent RL exploration method that suffers from scalability limitations. This establishes vMF-exp as a scalable and practical alternative to B-exp. While the primary focus of this paper is on the theoretical foundations of vMF-exp, extensive experiments on simulated data, real-world public data, and the successful deployment of vMF-exp on a music streaming service validated the scalability and practical relevance of the proposed method.

## Impact Statement

This paper introduces a scalable method for exploring large action sets in reinforcement learning problems, with applications including recommender systems. While our work is primarily methodological and theoretical, its integration into real-world systems, such as personalized music recommender systems, can influence user experience, engagement, and exposure to information. We highlight the potential for both positive outcomes (e.g., improved personalization and efficiency) and risks such as over-personalization or bias amplification if used without appropriate fairness and diversity constraints. We encourage careful deployment and further study when applying this method to sensitive domains involving social or behavioral data.

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

## Appendix

This appendix supplements the article "*Exploring Large Action Sets with Hyperspherical Embeddings using von Mises-Fisher Sampling*" and is organized as follows:

- Appendices A, B, C, and D provide detailed proofs and discussions for all theoretical results presented in the paper.

- Appendix E explores the connection between vMF-exp and Thompson Sampling.

- Appendix F explains practical methods for sampling from the vMF distribution.

- Appendix G presents the complete results of our Monte Carlo simulations.

- Appendix H details an additional experimental study using a large-scale, publicly available GloVe dataset.

- Appendix I highlights the successful large-scale deployment of vMF-exp in the private production system of the global music streaming service Deezer for large-scale music recommendation.

## A. Asymptotic Behavior of Boltzmann Exploration (Proof of Proposition 4.2)

We begin with the proof of Proposition 4.2 claiming that, in the setting of Section 4.1, we have:

$$
P_{\text{B-exp}}(a \mid n, d, V, \kappa) = \underbrace{\frac{f_{\text{vMF}}(A \mid V, \kappa)\mathcal{A}(\mathcal{S}^{d-1})}{n}}_{\text{denoted } P_0(a|n,d,V,\kappa)} + o\left(\frac{1}{n\sqrt{n}}\right),
\tag{22}
$$

with $f_{\text{vMF}}$ the probability density function (PDF) of the von Mises-Fisher (vMF) (Fisher, 1953) distribution:

$$
\forall A \in \mathcal{S}^{d-1}, f_{\text{vMF}}(A \mid V, \kappa) = C_d(\kappa)e^{\kappa\langle V,A\rangle},
\tag{23}
$$

where $\mathcal{A}(\mathcal{S}^{d-1})$ is the surface area of $\mathcal{S}^{d-1}$, the $d$-dimensional unit hypersphere, and $C_d(\kappa)$ is the normalizing constant.

*Proof.* By definition,

$$
\begin{aligned}
P_{\text{B-exp}}(a \mid n, d, V, \kappa) &= \mathbb{E}_{\mathcal{X}_n \sim \mathcal{U}(\mathcal{S}^{d-1})}\left[\frac{e^{\kappa\langle V,A\rangle}}{e^{\kappa\langle V,A\rangle} + \sum_{i=1}^n e^{\kappa\langle V,X_i\rangle}}\right] \\
&= \frac{e^{\kappa\langle V,A\rangle}}{n} \mathbb{E}_{\mathcal{X}_n \sim \mathcal{U}(\mathcal{S}^{d-1})}\left[\frac{1}{\frac{e^{\kappa\langle V,A\rangle}}{n} + \sum_{i=1}^n \frac{e^{\kappa\langle V,X_i\rangle}}{n}}\right] \\
&= \frac{e^{\kappa\langle V,A\rangle}}{n} \mathbb{E}_{\mathcal{X}_n \sim \mathcal{U}(\mathcal{S}^{d-1})}\left[\frac{1}{D_n}\right].
\end{aligned}
\tag{24}
$$

We use $D_n$ to denote the denominator of the expression inside the above expectation. $D_n$ is the empirical average of $n$ independent and identically distributed (i.i.d.) random variables (plus a constant). Therefore, by applying the *Central Limit Theorem (CLT)* (Fischer, 2011), we know that as $n$ grows it will be asymptotically distributed according to a Normal distribution with the following expectation:

$$
\begin{aligned}
\mathbb{E}_{\mathcal{X}_n \sim \mathcal{U}(\mathcal{S}^{d-1})}[D_n] &= \mathbb{E}_{\mathcal{X}_n \sim \mathcal{U}(\mathcal{S}^{d-1})}\left[\frac{e^{\kappa\langle V,A\rangle}}{n} + \sum_{i=1}^n \frac{e^{\kappa\langle V,X_i\rangle}}{n}\right] \\
&= \frac{e^{\kappa\langle V,A\rangle}}{n} + \mathbb{E}_{X \sim \mathcal{U}(\mathcal{S}^{d-1})}\left[e^{\kappa\langle V,X\rangle}\right].
\end{aligned}
\tag{25}
$$

Moreover, we have:

$$
\begin{aligned}
\mathbb{E}_{X \sim \mathcal{U}(\mathcal{S}^{d-1})}\left[e^{\kappa\langle V,X\rangle}\right] &= \int_{X \in \mathcal{S}^{d-1}} \frac{e^{\kappa\langle V,X\rangle}}{\mathcal{A}(\mathcal{S}^{d-1})}\, dX \\
&= \frac{1}{\mathcal{A}(\mathcal{S}^{d-1})C_d(\kappa)},
\end{aligned}
\tag{26}
$$

using the fact that $C_d(\kappa)$ is the normalizing constant of a vMF distribution, ensuring that its PDF (Equation (23)) sums to 1 when integrated on the unit hypersphere.

Let us define $\sigma = \mathrm{Var}_{X \sim \mathcal{U}(\mathcal{S}^{d-1})} \left[ e^{\kappa \langle V, X \rangle} \right]$ Although we do not need an explicit expression for $\sigma$, we know it is finite. Additionally, let $g : x \mapsto \dfrac{1}{x}$ be the inverse function. The CLT ensures that:

$$\sqrt{n}\Big[D_n - \frac{1}{\mathcal{A}(\mathcal{S}^{d-1})C_d(\kappa)}\Big] \xrightarrow{D} \mathcal{N}(0, \sigma^2), \tag{27}$$

where $\xrightarrow{D}$ denotes convergence in distribution (Jacod & Protter, 2004). Moreover, since $g$ is a differentiable function on $\mathbb{R}^*_+$, we use the *Delta method* (Oehlert, 1992) to infer that:

$$\sqrt{n}[g(D_n) - g(\frac{1}{\mathcal{A}(\mathcal{S}^{d-1})C_d(\kappa)})] \xrightarrow{D} \mathcal{N}(0, \sigma^2[g'(\frac{1}{\mathcal{A}(\mathcal{S}^{d-1})C_d(\kappa)})]^2). \tag{28}$$

Replacing $g$ and $g'$ by their respective values, we obtain:

$$\sqrt{n}\Big[\frac{1}{D_n} - C_d(\kappa)\mathcal{A}(\mathcal{S}^{d-1})\Big] \xrightarrow{D} \mathcal{N}(0, \sigma^2(\mathcal{A}(\mathcal{S}^{d-1})C_d(\kappa))^4). \tag{29}$$

Furthermore, recall that if a sequence $Z_1, Z_2, ...$ of random variables converges in distribution to a random variable $Z$, then for all bounded continuous function $\phi$, $\lim\limits_{n \to +\infty} \mathbb{E}\left[\phi(Z_n)\right] = \mathbb{E}\left[\phi(Z)\right]$ (Jacod & Protter, 2004). Since for every $n$ the random variable $Z_n = \sqrt{n}[\frac{1}{D_n} - C_d(\kappa)\mathcal{A}(\mathcal{S}^{d-1})]$ has bounded values, we can simply chose the identity function for $\phi$ to conclude that :

$$\lim_{n \to +\infty} \mathbb{E}_{X_n \sim \mathcal{U}(\mathcal{S}^{d-1})} \left[\sqrt{n}[\frac{1}{D_n} - C_d(\kappa)\mathcal{A}(\mathcal{S}^{d-1})]\right] = 0, \tag{30}$$

which is equivalent to:

$$\mathbb{E}_{X_n \sim \mathcal{U}(\mathcal{S}^{d-1})} \left[\frac{1}{D_n}\right] = C_d(\kappa)\mathcal{A}(\mathcal{S}^{d-1}) + o(\frac{1}{\sqrt{n}}). \tag{31}$$

Finally, by multiplying Equation (31) by $\frac{e^{\kappa \langle V, A \rangle}}{n}$, we obtain Equation (22), concluding the proof. $\qquad\square$

# B. Asymptotic Behavior of vMF Exploration in $d = 2$ dimensions (Proof of Proposition 4.3, Part 1)

We now prove Proposition 4.3 when $d = 2$. In 2 dimensions, the vMF distribution takes the special form of the von Mises (vM) distribution (Mardia & Jupp, 2009) which, instead of describing the distribution of the dot product between $V$ and $\tilde{V}$, describes the distribution of their angle $\theta$. The PDF of a von Mises distribution is defined as follows:

$$\forall \theta \in [-\pi, \pi], f_{vM}(\theta \mid \kappa) = \frac{e^{\kappa \cos(\theta)}}{2\pi I_0(\kappa)}. \tag{32}$$

Let us define $\theta_0$ as the angle between $V$ and $A$. In this section, we prove that:

$$P_{\text{vMF-exp}}(A \mid n, d = 2, \kappa) = \frac{e^{\kappa \cos(\theta_0)}}{n I_0(\kappa)} + \mathcal{O}(\frac{1}{n^2}). \tag{33}$$

*Proof.* By definition,

$$P_{\text{vMF-exp}}(A \mid n, d = 2, \kappa) = \mathbb{E}_{\mathcal{X}_n \sim \mathcal{U}(\mathcal{S}^1)} \left[ \mathbb{P}(\tilde{V} \in \mathcal{S}_{\text{Voronoï}}(A \mid \mathcal{X}_{n+1})) \right], \tag{34}$$

where $\mathcal{S}_{\text{Voronoï}}(X_i \mid \mathcal{X}_n) = \{\tilde{V} \in \mathcal{S}^{d-1}, \forall j \in \mathcal{I}_n, \langle \tilde{V}, X_i \rangle \geq \langle \tilde{V}, X_j \rangle \}$. Let us call $\mathcal{Y}_n = \{Y_i\}$ the result of the permutation of the indices of $\mathcal{X}_n$ such that the (signed) angles $\beta_i$ between $A$ and $Y_i$ are sorted in increasing order. Since the $\{X_i\}$ are i.i.d. and uniformly distributed on the circle, then the angles between $A$ and the $\{X_i\}$ are i.i.d. and uniformly distributed on $[0, 2\pi]$. Therefore, the set $\{\beta_i\}$ is the set of the order statistics of $n$ i.i.d. random variables uniformly distributed on $[0, 2\pi]$. Consequently, the set $\{\frac{\beta_i}{2\pi}\}$ is the set of the order statistics of $n$ i.i.d. random variables uniformly distributed on $[0, 1]$, which are known to follow Beta distributions (Gentle, 2009) defined as follows:

$$\forall 1 \leq i \leq n, \frac{\beta_i}{2\pi} \sim \text{Beta}(i, n + 1 - i). \tag{35}$$

As a consequence, we have:

$$\mathbb{E}[\beta_1] = \frac{2\pi}{n + 1}, \tag{36}$$

$$\mathbb{E}[\beta_n] = \frac{2\pi n}{n + 1}, \tag{37}$$

$$\text{Var}[\beta_1] = \text{Var}[\beta_n] = \frac{4\pi^2 n}{(n + 1)^2 (n + 2)}. \tag{38}$$

Moreover, for given values of $Y_i$, we can see from Figure 4 that, in 2 dimensions, Voronoï cells are arcs of the circle and are delimited by perpendicular bisectors of two neighboring points. Specifically, the Voronoï cell of $A$ is delimited by the perpendicular bisector of A and $Y_1$ on one side, and the perpendicular bisector of A and $Y_n$ on the other side. By denoting $\theta$ the (signed) angle between $V$ and $\tilde{V}$, we have:

$$\mathbb{P}\left(\tilde{V} \in \mathcal{S}_{\text{Voronoï}}(A \mid \mathcal{X}_{n+1})\right) = \mathbb{P}\left(\theta \in [\theta_0 + \frac{\beta_n - 2\pi}{2}, \theta_0 + \frac{\beta_1}{2}] \mid \theta \sim \text{vM}(0, \kappa), \beta_1, \beta_n\right)$$
$$= \int_{\theta = \theta_0 + \frac{\beta_n - 2\pi}{2}}^{\theta_0 + \frac{\beta_1}{2}} f_{vM}(\theta \mid \kappa) \, d\theta. \tag{39}$$

Therefore:

$$P_{\text{vMF-exp}}(A \mid n, d = 2, \kappa) = \mathbb{E}_{\beta_1, \beta_n} \left[ \int_{\theta = \theta_0 + \frac{\beta_n - 2\pi}{2}}^{\theta_0 + \frac{\beta_1}{2}} f_{vM}(\theta \mid \kappa) \, d\theta \right]. \tag{40}$$

To get an asymptotic expression of the probability that $\theta$ lies between the considered bounds, we can first notice that as $n$ grows, $\beta_1$ will approach 0 and $\beta_n$ will approach $2\pi$. This means that the integral we need to compute will have very narrow

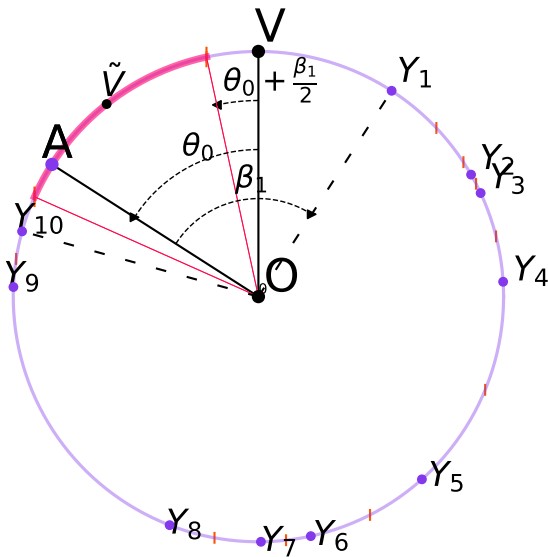

*Figure 4.* For $d = 2$: vMF-exp explores the action $A$ when $\tilde{V}$ lies in its Voronoï cell, shown in red.

bounds centered on $\theta_0$, and so we can leverage the Taylor series expansion (Abramowitz & Stegun, 1948) of $f_{\text{vM}}$ around $\theta_0$ and obtain:

$$f_{\text{vM}}(\theta \mid \kappa) = f_{\text{vM}}(\theta_0 \mid \kappa) + R_0(\theta), \tag{41}$$

where $R_0(\theta) = \sum_{i=1}^{\infty} \frac{f_{\text{vM}}^{(i)}(\theta_0 \mid \kappa)}{i!} (\theta - \theta_0)^i$ is the zero order remainder term of the Taylor series expansion of $f_{\text{vM}}$ near $\theta_0$.

We can now estimate the portion of the integral of Equation (40) corresponding to each term of the expansion separately, and show that when $n$ becomes large:

- the zero-order term gives a probability of selecting $A$ that is the same as the asymptotic behavior of B-exp:
  $\mathbb{E}_{\beta_1, \beta_n} \left[ \int_{\theta = \theta_0 + \frac{\beta_n - 2\pi}{2}}^{\theta_0 + \frac{\beta_1}{2}} f_{\text{vM}}(\theta_0 \mid \kappa) \, d\theta \right] = \frac{e^{\kappa \cos(\theta_0)}}{n I_0(\kappa)} + \mathcal{O}(\frac{1}{n^2})$.

- the expectation of the remainder term is bounded by a $\frac{1}{n^2}$ term: $\mathbb{E}_{\beta_1, \beta_n} \left[ \int_{\theta = \theta_0 + \frac{\beta_n - 2\pi}{2}}^{\theta_0 + \frac{\beta_1}{2}} R_0(\theta) \, d\theta \right] = \mathcal{O}(\frac{1}{n^2})$.

## B.1. Zero-Order Estimate

Let us study the zero-order approximation of $f_{\text{vM}}(\theta \mid \kappa)$ near $\theta_0$:

$$
\begin{aligned}
\mathbb{E}_{\beta_1,\beta_n}\left[\int_{\theta=\theta_0+\frac{\beta_n-2\pi}{2}}^{\theta_0+\frac{\beta_1}{2}} f_{\text{vM}}(\theta_0 \mid \kappa)\,\mathrm{d}\theta\right] &= \mathbb{E}_{\beta_1,\beta_n}\left[f_{\text{vM}}(\theta_0 \mid \kappa)(\theta_0 + \frac{\beta_1}{2} - (\theta_0 + \frac{\beta_n-2\pi}{2}))\right] \\
&= \mathbb{E}_{\beta_1,\beta_n}\left[f_{\text{vM}}(\theta_0 \mid \kappa)(\pi - \frac{\beta_n-\beta_1}{2})\right] \\
&= \pi f_{\text{vM}}(\theta_0 \mid \kappa)\,\mathbb{E}_{\beta_1,\beta_n}\left[1 - \frac{\beta_n-\beta_1}{2\pi}\right] \\
&= \frac{e^{\kappa\cos(\theta)}}{2I_0(\kappa)}(1 - \mathbb{E}_{\beta_1,\beta_n}\left[\frac{\beta_n}{2\pi}\right] + \mathbb{E}_{\beta_1,\beta_n}\left[\frac{\beta_1}{2\pi}\right]) \\
&= \frac{e^{\kappa\cos(\theta_0)}}{2I_0(\kappa)}\frac{n+1-n+1}{n+1} \\
&= \frac{e^{\kappa\cos(\theta_0)}}{2I_0(\kappa)}\frac{2}{n+1} \\
&= \frac{e^{\kappa\cos(\theta_0)}}{(n+1)I_0(\kappa)} \\
&= \frac{e^{\kappa\cos(\theta_0)}}{nI_0(\kappa)} - \frac{e^{\kappa\cos(\theta_0)}}{n(n+1)I_0(\kappa)} \\
&= \frac{e^{\kappa\cos(\theta_0)}}{nI_0(\kappa)} + \mathcal{O}(\frac{1}{n^2}).
\end{aligned}
\tag{42}
$$

This proves that, asymptotically, the contribution of the zero-order term of $f_{\text{vM}}$ to the probability of selecting $A$ is equal to the probability of selecting $A$ using B-exp with the same $\kappa$ value.

To understand how fast vMF-exp reaches its asymptotic behavior, we now need to study $R_0(\theta)$, the remainder of the Taylor series expansion of $f_{\text{VM}}$ around $\theta_0$.

## B.2. Bounding of the Remainder Term

We start by computing the first derivative of $f_{\text{vM}}$:

$$
\forall \theta \in [0, 2\pi], |f'_{\text{vM}}(\theta \mid \kappa)| = \frac{|\sin(\theta)|\kappa e^{\kappa\cos(\theta)}}{I_0(\kappa)},
\tag{43}
$$

which is bounded[4] on $[0, 2\pi]$ by $M = \frac{\kappa e^{\kappa}}{I_0(\kappa)}$. According to the Taylor-Lagrange inequality (Abramowitz & Stegun, 1948), this in turn bounds the remainder term as follows:

$$
\forall \theta \in [0, 2\pi], |R_0(\theta)| \leq M|\theta - \theta_0|.
\tag{44}
$$

---

[4]We note that a tighter bound could be found by studying the second derivative, but will not be necessary for the purpose of this proof.

In particular, this inequality holds for every $\theta \in [\theta_0 + \frac{\beta_n - 2\pi}{2}, \theta_0 + \frac{\beta_1}{2}]$, and so:

$$
\begin{aligned}
\int_{\theta=\theta_0+\frac{\beta_n-2\pi}{2}}^{\theta_0+\frac{\beta_1}{2}} |R_0(\theta)|\, \mathrm{d}\theta &\leq \int_{\theta=\theta_0+\frac{\beta_n-2\pi}{2}}^{\theta_0+\frac{\beta_1}{2}} M|\theta - \theta_0|\, \mathrm{d}\theta \\
&= \int_{\theta=\theta_0}^{\theta_0+\frac{\beta_1}{2}} M(\theta - \theta_0)\, \mathrm{d}\theta + \int_{\theta=\theta_0+\frac{\beta_n-2\pi}{2}}^{\theta_0} M(\theta_0 - \theta)\, \mathrm{d}\theta \\
&= \int_{\theta=0}^{\frac{\beta_1}{2}} M\theta\, \mathrm{d}\theta - \int_{\theta=\frac{\beta_n-2\pi}{2}}^{0} M\theta\, \mathrm{d}\theta \\
&= M\frac{\beta_1^2 + (\beta_n - 2\pi)^2}{8}.
\end{aligned}
\tag{45}
$$

The above inequality holds when considering the expected values over uniformly distributed $X_i$:

$$
\begin{aligned}
\mathbb{E}_{\beta_1,\beta_n}\left[\int_{\theta=\theta_0+\frac{\beta_n-2\pi}{2}}^{\theta_0+\frac{\beta_1}{2}} |R_0(\theta)|\, \mathrm{d}\theta\right] &\leq M\frac{\mathbb{E}_{\beta_1,\beta_n}\left[\beta_1^2\right] + \mathbb{E}_{\beta_1,\beta_n}\left[(\beta_n - 2\pi)^2\right]}{8} \\
&= M\frac{\mathrm{Var}_{\beta_1,\beta_n}\left[\beta_1\right] + (\mathbb{E}_{\beta_1,\beta_n}\left[\beta_1\right])^2 + \mathrm{Var}_{\beta_1,\beta_n}\left[(\beta_n - 2\pi)\right] + (\mathbb{E}_{\beta_1,\beta_n}\left[\beta_n - 2\pi\right])^2}{8} \\
&= \frac{M}{8}\left(\frac{2 \times 4\pi^2 n}{(n+1)^2(n+2)} + \frac{2 \times 4\pi^2}{(n+1)^2}\right) \\
&= \frac{M\pi^2}{(n+1)(n+2)} \\
&= \mathcal{O}(\frac{1}{n^2}).
\end{aligned}
\tag{46}
$$

Since $\left|\mathbb{E}_{\beta_1,\beta_n}\left[\int_{\theta=\theta_0+\frac{\beta_n-2\pi}{2}}^{\theta_0+\frac{\beta_1}{2}} R_0(\theta)\, \mathrm{d}\theta\right]\right| \leq \mathbb{E}_{\beta_1,\beta_n}\left[\int_{\theta=\theta_0+\frac{\beta_n-2\pi}{2}}^{\theta_0+\frac{\beta_1}{2}} |R_0(\theta)|\, \mathrm{d}\theta\right]$, we have shown:

$$
\mathbb{E}_{\beta_1,\beta_n}\left[\int_{\theta=\theta_0+\frac{\beta_n-2\pi}{2}}^{\theta_0+\frac{\beta_1}{2}} R_0(\theta)\, \mathrm{d}\theta\right] = \mathcal{O}(\frac{1}{n^2}).
\tag{47}
$$

In summary, when combining the asymptotic behavior of the zero-order term and the remainder term, we conclude that when $d = 2$ we have:

$$
P_{\text{vMF-exp}}(A \mid n, d = 2, \kappa) = \frac{e^{\kappa \cos(\theta_0)}}{nI_0(\kappa)} + \mathcal{O}(\frac{1}{n^2}).
\tag{48}
$$

This proves Proposition 4.3 when $d = 2$. Note that, comparing the asymptotic expressions for $P_{\text{B-exp}}(A \mid n, d = 2, \kappa)$ and $P_{\text{vMF-exp}}(A \mid n, d = 2, \kappa)$, also gives us a proof for Proposition 4.1 when $d = 2$. $\qquad\square$

## C. Asymptotic Behavior of vMF Exploration in $d > 2$ dimensions (Proofs of Proposition 4.3, Part 2, and of Proposition 4.4)

We now prove Proposition 4.3 when $d > 2$, starting with a series of intermediary lemmas. We subsequently justify the approximate expression of Proposition 4.4.

### C.1. Intermediary Lemmas

We introduce a series of lemmas regarding the properties of the Voronoï cell of $A$ when $\mathcal{X}_n \sim \mathcal{U}^{d-1}$. We recall that, for a given set of embedding vectors $\mathcal{X}_n$, we use the notation $\mathcal{X}_{n+1} = \mathcal{X}_n \cup \{A\}$.

**Lemma C.1.** *Let $d \in \mathbb{N}, d \geq 2$, $A \in \mathcal{S}^{d-1}$ and $n \in \mathbb{N}^*$. As before, let $\mathcal{A}(\mathcal{S}^{d-1})$ denote the surface area of $\mathcal{S}^{d-1}$. Then:*

$$\mathbb{E}_{\mathcal{X}_n \sim \mathcal{U}(\mathcal{S}^{d-1})}\Big[\mathcal{A}(\mathcal{S}_{Voronoi}(A \mid \mathcal{X}_{n+1}))\Big] = \frac{\mathcal{A}(\mathcal{S}^{d-1})}{n+1}. \tag{49}$$

*Proof.* To compute this expectation, one can notice that:

$$\mathbb{E}_{\mathcal{X}_n \sim \mathcal{U}(\mathcal{S}^{d-1})}\Big[\mathcal{A}(\mathcal{S}_{\text{Voronoï}}(A \mid \mathcal{X}_{n+1}))\Big] = \mathbb{E}_{\mathcal{X}_{n+1} \sim \mathcal{U}(\mathcal{S}^{d-1})}\Big[\mathcal{A}(\mathcal{S}_{\text{Voronoï}}(X_{n+1} \mid \mathcal{X}_{n+1})) \mid X_{n+1} = A\Big]. \tag{50}$$

Indeed, considering that $A$ is known is equivalent to considering $A$ as a random vector $X_{n+1} \sim \mathcal{U}(\mathcal{S}^{d-1})$ with the constraint $X_{n+1} = A$. We will now show that the right part of Equation (50) is actually independent of the value of $A$.

Consider any point $A' \in \mathcal{S}^{d-1}$. One can always define a (not necessarily unique) rotation $R_{A,A'}$ such that $R_{A,A'}(A) = A'$. Since rotations preserve inner products, they also preserve areas of Voronoi cells, which means that for a given set of vectors $\mathcal{X}_{n+1}$, we have:

$$\mathcal{A}\Big(\mathcal{S}_{\text{Voronoï}}(X_{n+1} \mid \mathcal{X}_{n+1})\Big) = \mathcal{A}\Big(\mathcal{S}_{\text{Voronoï}}(R_{A,A'}(X_{n+1}) \mid R_{A,A'}(\mathcal{X}_{n+1}))\Big). \tag{51}$$

Moreover, the image of the rotation of a random vector uniformly distributed on the hypersphere is also uniformly distributed, which means that:

$$\mathcal{X}_{n+1} \sim \mathcal{U}(\mathcal{S}^{d-1}) \Leftrightarrow R_{A,A'}(\mathcal{X}_{n+1}) \sim \mathcal{U}(\mathcal{S}^{d-1}). \tag{52}$$

Therefore:

$$
\begin{aligned}
&\mathbb{E}_{\mathcal{X}_{n+1} \sim \mathcal{U}(\mathcal{S}^{d-1})}\Big[\mathcal{A}(\mathcal{S}_{\text{Voronoï}}(X_{n+1} \mid \mathcal{X}_{n+1})) \mid X_{n+1} = A\Big] \\
=&\mathbb{E}_{\mathcal{X}_{n+1} \sim \mathcal{U}(\mathcal{S}^{d-1})}\Big[\mathcal{A}(\mathcal{S}_{\text{Voronoï}}(R_{A,A'}(X_{n+1}) \mid R_{A,A'}(\mathcal{X}_{n+1}))) \mid X_{n+1} = A\Big] \\
=&\mathbb{E}_{R_{A,A'}(\mathcal{X}_{n+1}) \sim \mathcal{U}(\mathcal{S}^{d-1})}\Big[\mathcal{A}(\mathcal{S}_{\text{Voronoï}}(R_{A,A'}(X_{n+1}) \mid R_{A,A'}(\mathcal{X}_{n+1}))) \mid R_{A,A'}(X_{n+1}) = A'\Big] \\
=&\mathbb{E}_{R_{A,A'}(\mathcal{X}_n) \sim \mathcal{U}(\mathcal{S}^{d-1})}\Big[\mathcal{A}(\mathcal{S}_{\text{Voronoï}}(A' \mid R_{A,A'}(\mathcal{X}_n)))\Big] \\
=&\mathbb{E}_{\mathcal{X}_n \sim \mathcal{U}(\mathcal{S}^{d-1})}\Big[\mathcal{A}(\mathcal{S}_{\text{Voronoï}}(A' \mid \mathcal{X}_n))\Big].
\end{aligned}
\tag{53}
$$

This result proves that $\mathbb{E}_{\mathcal{X}_n \sim \mathcal{U}(\mathcal{S}^{d-1})}[\mathcal{A}(\mathcal{S}_{\text{Voronoï}}(A \mid \mathcal{X}_{n+1}))]$ is independent of $A$. Then, we use this information along with Equation (50) to obtain:

$$\mathbb{E}_{\mathcal{X}_n \sim \mathcal{U}(\mathcal{S}^{d-1})}\Big[\mathcal{A}(\mathcal{S}_{\text{Voronoï}}(A \mid \mathcal{X}_{n+1}))\Big] = \mathbb{E}_{\mathcal{X}_{n+1} \sim \mathcal{U}(\mathcal{S}^{d-1})}\Big[\mathcal{A}(\mathcal{S}_{\text{Voronoï}}(X_{n+1} \mid \mathcal{X}_{n+1}))\Big]. \tag{54}$$

Since $\sum_{i=1}^{n+1} \mathcal{A}(\mathcal{S}_{\text{Voronoï}}(X_i \mid \mathcal{X}_{n+1})) = \mathcal{A}(\mathcal{S}^{d-1})$ (Du et al., 1999; 2010) and the $X_i$ are i.i.d., we derive:

$$\mathbb{E}_{\mathcal{X}_{n+1} \sim \mathcal{U}(\mathcal{S}^{d-1})}\Big[\mathcal{A}(\mathcal{S}_{\text{Voronoï}}(X_{n+1} \mid \mathcal{X}_{n+1}))\Big] = \frac{\mathcal{A}(\mathcal{S}^{d-1})}{n+1}. \tag{55}$$

Combining Equations (50) with Equation (55) leads to Equation (49), concluding the proof. $\square$

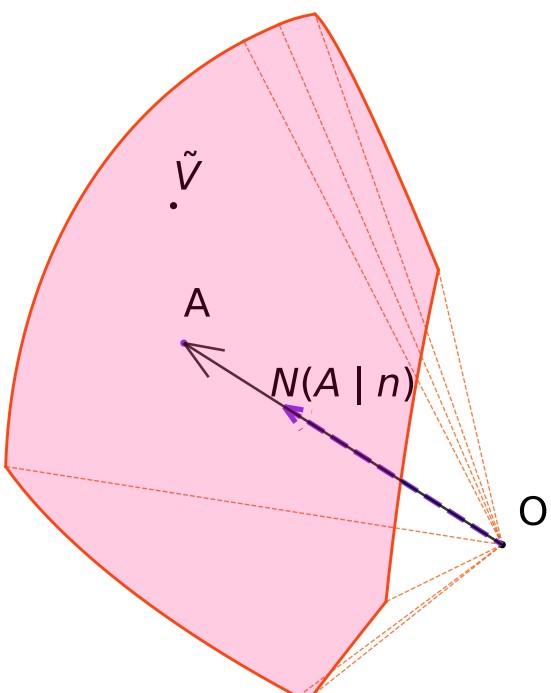

*Figure 5.* The Voronoï cell of $A$, $\mathcal{S}_{\text{Voronoï}}(A \mid \mathcal{X}_{n+1})$, along with the average normal vector of the cell $N(A \mid n)$. On expectation, $(A \mid n)$ and $A$ are collinear.

**Lemma C.2.** *Let $d \in \mathbb{N}, d \geq 2$, $A \in \mathcal{S}^{d-1}$ and $n \in \mathbb{N}^*$. Then:*

$$\exists \lambda \in \mathbb{R}, \mathbb{E}_{\mathcal{X}_n \sim \mathcal{U}(\mathcal{S}^{d-1})}\left[\int_{\tilde{V} \in \mathcal{S}_{\text{Voronoï}}(A|\mathcal{X}_{n+1})} \tilde{V} \, \mathrm{d}\tilde{V}\right] = \lambda A. \tag{56}$$

*Proof.* We want to prove that the average normal vector of the Voronoï cell of $A$ and $A$ are collinear, as illustrated in Figure 5. To do so, we will show that this average normal vector is invariant to any rotation around $A$. For every $\theta \in [0, 2\pi]$, we define $R_{A,\theta}$ as the rotation around $A$ of the angle $\theta$. As discussed in the proof of Lemma C.1, $\mathcal{X}_n \sim \mathcal{U}(\mathcal{S}^{d-1}) \Leftrightarrow R_{A,\theta}(\mathcal{X}_n) \sim \mathcal{U}(\mathcal{S}^{d-1})$. Moreover, $R_{A,\theta}(A) = A$. Let us denote:

$$N(A \mid n) = \mathbb{E}_{\mathcal{X}_n \sim \mathcal{U}(\mathcal{S}^{d-1})}\left[\int_{\tilde{V} \in \mathcal{S}_{\text{Voronoï}}(A|\mathcal{X}_{n+1})} \tilde{V} \, \mathrm{d}\tilde{V}\right], \tag{57}$$

the expected normal vector of the Voronoï cell of $A$. Its image by the rotation $R_{A,\theta}$ verifies:

$$\begin{aligned}
R_{A,\theta}(N(A \mid n)) &= R_{A,\theta}\left(\mathbb{E}_{\mathcal{X}_n \sim \mathcal{U}(\mathcal{S}^{d-1})}\left[\int_{\tilde{V} \in \mathcal{S}_{\text{Voronoï}}(A|\mathcal{X}_{n+1})} \tilde{V} \, \mathrm{d}\tilde{V}\right]\right) \\
&= \mathbb{E}_{\mathcal{X}_n \sim \mathcal{U}(\mathcal{S}^{d-1})}\left[\int_{\tilde{V} \in \mathcal{S}_{\text{Voronoï}}(R_{A,\theta}(A)|R_{A,\theta}(\mathcal{X}_{n+1}))} \tilde{V} \, \mathrm{d}\tilde{V}\right] \\
&= \mathbb{E}_{R_{A,\theta}(\mathcal{X}_n) \sim \mathcal{U}(\mathcal{S}^{d-1})}\left[\int_{\tilde{V} \in \mathcal{S}_{\text{Voronoï}}(A|R_{A,\theta}(\mathcal{X}_{n+1}))} \tilde{V} \, \mathrm{d}\tilde{V}\right] \\
&= N(A \mid n).
\end{aligned} \tag{58}$$

This proves that $N(A \mid n)$ and $A$ are collinear. $\qquad\square$

**Lemma C.3.** *With the same hypotheses as Lemma C.2:*

$$\lambda = \frac{\mathcal{A}(\mathcal{S}^{d-1})}{n+1} \mathbb{E}_{\mathcal{X}_n \sim \mathcal{U}(\mathcal{S}^{d-1}), \tilde{V} \sim \mathcal{U}(\mathcal{S}^{d-1})}\left[\max_i \langle \tilde{V}, X_i \rangle\right]. \tag{59}$$

*Proof.* $\lambda$ is defined as follows:

$$\lambda A = \mathbb{E}_{\mathcal{X}_n \sim \mathcal{U}(\mathcal{S}^{d-1})} \Big[ \int_{\tilde{V} \in \mathcal{S}_{\text{Voronoï}}(A|\mathcal{X}_{n+1})} \tilde{V} \, d\tilde{V} \Big]$$

$$\implies \langle \lambda A, A \rangle = \langle \mathbb{E}_{\mathcal{X}_n \sim \mathcal{U}(\mathcal{S}^{d-1})} \Big[ \int_{\tilde{V} \in \mathcal{S}_{\text{Voronoï}}(A|\mathcal{X}_{n+1})} \tilde{V} \, d\tilde{V} \Big], A \rangle$$

$$\Leftrightarrow \quad \lambda = \mathbb{E}_{\mathcal{X}_n \sim \mathcal{U}(\mathcal{S}^{d-1})} \Big[ \int_{\tilde{V} \in \mathcal{S}_{\text{Voronoï}}(A|\mathcal{X}_{n+1})} \langle \tilde{V}, A \rangle \, d\tilde{V} \Big] \tag{60}$$

$$\Leftrightarrow \quad \lambda = \mathbb{E}_{\mathcal{X}_{n+1} \sim \mathcal{U}(\mathcal{S}^{d-1})} \Big[ \int_{\tilde{V} \in \mathcal{S}_{\text{Voronoï}}(X_{n+1}|\mathcal{X}_{n+1})} \langle \tilde{V}, X_{n+1} \rangle \, d\tilde{V} \mid X_{n+1} = A \Big]$$

$$\Leftrightarrow \quad \lambda = \mathbb{E}_{\mathcal{X}_{n+1} \sim \mathcal{U}(\mathcal{S}^{d-1})} \Big[ \int_{\tilde{V} \in \mathcal{S}_{\text{Voronoï}}(X_{n+1}|\mathcal{X}_{n+1})} \max_i \langle \tilde{V}, X_i \rangle \, d\tilde{V} \mid X_{n+1} = A \Big].$$

Moreover, as done in the proof of Lemma C.1, we can leverage the invariance by any rotation of the above expression to infer that the conditional expectation is actually independent of $A$:

$$\lambda = \mathbb{E}_{\mathcal{X}_{n+1} \sim \mathcal{U}(\mathcal{S}^{d-1})} \Big[ \int_{\tilde{V} \in \mathcal{S}_{\text{Voronoï}}(X_{n+1}|\mathcal{X}_{n+1})} \max_i \langle \tilde{V}, X_i \rangle \, d\tilde{V} \Big]. \tag{61}$$

Since, in the above equation, $X_{n+1}$ has the same distribution as every element of $\mathcal{X}_{n+1}$, a similar expression for $\lambda$ can be found using each $\mathcal{X}_{n+1}$ element. By summing them together, we obtain:

$$(n+1)\lambda = \sum_{j=1}^{n+1} \mathbb{E}_{\mathcal{X}_{n+1} \sim \mathcal{U}(\mathcal{S}^{d-1})} \Big[ \int_{\tilde{V} \in \mathcal{S}_{\text{Voronoï}}(X_j|\mathcal{X}_{n+1})} \max_i \langle \tilde{V}, X_i \rangle \, d\tilde{V} \Big]$$

$$= \mathbb{E}_{\mathcal{X}_{n+1} \sim \mathcal{U}(\mathcal{S}^{d-1})} \Big[ \sum_{j=1}^{n+1} \int_{\tilde{V} \in \mathcal{S}_{\text{Voronoï}}(X_j|\mathcal{X}_{n+1})} \max_i \langle \tilde{V}, X_i \rangle \, d\tilde{V} \Big]$$

$$= \mathbb{E}_{\mathcal{X}_{n+1} \sim \mathcal{U}(\mathcal{S}^{d-1})} \Big[ \int_{\tilde{V} \in \mathcal{S}^{d-1}} \max_i \langle \tilde{V}, X_i \rangle \, d\tilde{V} \Big] \tag{62}$$

$$= \mathbb{E}_{\mathcal{X}_{n+1} \sim \mathcal{U}(\mathcal{S}^{d-1})} \Big[ \int_{\tilde{V} \in \mathcal{S}^{d-1}} \frac{\mathcal{A}(\mathcal{S}^{d-1}) \max_i \langle \tilde{V}, X_i \rangle}{\mathcal{A}(\mathcal{S}^{d-1})} \, d\tilde{V} \Big]$$

$$= \mathcal{A}(\mathcal{S}^{d-1}) \, \mathbb{E}_{\mathcal{X}_{n+1} \sim \mathcal{U}(\mathcal{S}^{d-1})} \Big[ \mathbb{E}_{\tilde{V} \sim \mathcal{U}(\mathcal{S}^{d-1})} [\max_i \langle \tilde{V}, X_i \rangle] \Big],$$

which proves the lemma. $\square$

The last two lemmas are useful to describe the distribution of $\max_i \langle \tilde{V}, X_i \rangle$ when $\tilde{V}$ is fixed, $\mathcal{X}_{n+1} \sim \mathcal{U}(\mathcal{S}^{d-1})$, and $n$ is large.

**Lemma C.4.** *Let* $B : (z_1, z_2) \mapsto \int_0^1 t^{z_1-1}(1-t)^{z_2-1} \, dt$ *denote the Beta function. Let* $d \geq 3$, $\tilde{V} \in \mathcal{S}^{d-1}$ *and* $X$ *be a random vector with* $X \sim \mathcal{U}(\mathcal{S}^{d-1})$. *Let* $F_{radial}$ *be the cumulative distribution function (CDF) of* $\langle \tilde{V}, X \rangle$. *The Taylor series expansion of* $F_{radial}$ *near 1 is:*

$$F_{radial}(t) = 1 - \frac{2^{\frac{d-1}{2}}}{(d-1)B(\frac{1}{2}, \frac{d-1}{2})} (1-t)^{\frac{d-1}{2}} + o((1-t)^{\frac{d-1}{2}}). \tag{63}$$

*Proof.* The distribution of $\langle \tilde{V}, X \rangle$ has been studied in directional statistics (Mardia & Jupp, 2009). Its PDF is known to be:

$$
\begin{aligned}
f_{\text{radial}}(t) &= \frac{(1-t^2)^{\frac{d-1}{2}-1}}{B(\frac{1}{2}, \frac{d-1}{2})} \\
&= \frac{(1-t)^{\frac{d-1}{2}-1}(1+t)^{\frac{d-1}{2}-1}}{B(\frac{1}{2}, \frac{d-1}{2})} \\
&= \frac{(1-t)^{\frac{d-1}{2}-1}(2-(1-t))^{\frac{d-1}{2}-1}}{B(\frac{1}{2}, \frac{d-1}{2})} \\
&= \frac{2^{\frac{d-1}{2}-1}(1-t)^{\frac{d-1}{2}-1}(1-\frac{(1-t)}{2})^{\frac{d-1}{2}-1}}{B(\frac{1}{2}, \frac{d-1}{2})} \\
&= \frac{2^{\frac{d-1}{2}-1}(1-t)^{\frac{d-1}{2}-1}}{B(\frac{1}{2}, \frac{d-1}{2})} \left( \sum_{i=0}^{=\infty} \binom{\frac{d-1}{2}-1}{i} \left(\frac{1-t}{2}\right)^i \right).
\end{aligned}
\tag{64}
$$

The last line above was obtained using Newton's generalized binomial theorem for real exponent (Coolidge, 1949). It involves the term $\binom{\frac{d-1}{2}-1}{i} = \frac{(\frac{d-1}{2}-1)_i}{i!}$ with $(\cdot)_i$ the Pochhammer symbol used to designate a falling factorial (Abramowitz & Stegun, 1948). We have obtained an expression of $f_{\text{radial}}$ involving an infinite weighted sum of powers of $(1-t)$ with exponents greater or equal to 0 since $d \geq 3$. Therefore, by uniqueness of the Taylor polynomial, we derive that the Taylor series expansion of $f_{\text{radial}}$ near 1 is:

$$
f_{\text{radial}}(t) = \frac{2^{\frac{d-1}{2}-1}(1-t)^{\frac{d-1}{2}-1}}{B(\frac{1}{2}, \frac{d-1}{2})} + o((1-t)^{\frac{d-1}{2}-1}).
\tag{65}
$$

Since by definition $F_{\text{radial}}$ is the primitive of $f_{\text{radial}}$ on $[-1, 1]$ and that $F_{\text{radial}}(1) = 1$, we can integrate the above equation to get:

$$
\begin{aligned}
F_{\text{radial}}(t) &= 1 - \frac{2}{d-1} \frac{2^{\frac{d-1}{2}-1}(1-t)^{\frac{d-1}{2}}}{B(\frac{1}{2}, \frac{d-1}{2})} + o((1-t)^{\frac{d-1}{2}}) \\
&= 1 - \frac{2^{\frac{d-1}{2}}(1-t)^{\frac{d-1}{2}}}{(d-1)B(\frac{1}{2}, \frac{d-1}{2})} + o((1-t)^{\frac{d-1}{2}}).
\end{aligned}
\tag{66}
$$

Since this is exactly the Equation (63), this completes the proof. $\square$

**Lemma C.5.** *Let $d \geq 3$, $\tilde{V} \in \mathcal{S}^{d-1}$ and let $F_{radial}$ be defined as in Lemma C.4. For $n \in \mathbb{N}^*$, let $\mathcal{X}_n \sim \mathcal{U}(\mathcal{S}^{d-1})$ be a set of $n$ i.i.d. random vectors uniformly distributed on $\mathcal{S}^{d-1}$, and let $F_n$ be the CDF of $\max_i \langle \tilde{V}, X_i \rangle$. Then, for $u \in [-1, 1]$:*

$$
\lim_{n \to +\infty} F_n(a_n u + b_n) = e^{(-(1+\gamma u)^{\frac{-1}{\gamma}})},
\tag{67}
$$

*where $\gamma = -\frac{2}{d-1}$, $a_n = \frac{1}{2}\left(\frac{(d-1)B(\frac{1}{2}, \frac{d-1}{2})}{n}\right)^{\frac{2}{d-1}}$ with $B$ the Beta function, and $b_n = 1 - \frac{2}{d-1}a_n$.*

*Proof.* The proof relies on the Fisher–Tippett–Gnedenko theorem (Gnedenko, 1943) which states that if there exists a couple of sequences $a_n$ and $b_n$ such that the left term of Equation (67) converges, then its limit should be the CDF of a Generalized Extreme Value distribution (GEV) with shape parameter $\gamma$, which is the right term of Equation (67). Theorem 5 of Gnedenko (1943) provides a necessary and sufficient convergence condition for a random variable with maximal value $x_{\max}$ and CDF F, provided that $\gamma < 0$:

$$
\lim_{t \to 0^+} \frac{1 - F(x_{\max} - ut)}{1 - F(x_{\max} - t)} = u^{\left(\frac{-1}{\gamma}\right)} \text{ for all } u > 0.
\tag{68}
$$

Recall that Lemma C.4 gives us the Taylor expansion of $F_{\text{radial}}$ near $1$ : $F_{\text{radial}}(t) = 1 - K(1-t)^{\frac{d-1}{2}} + o((1-t)^{\frac{d-1}{2}})$ with $K = \frac{2^{\frac{d-1}{2}}}{(d-1)B(\frac{1}{2}, \frac{d-1}{2})}$. Knowing that $x_{\max} = 1$, we obtain that, $\forall u > 0$:

$$
\begin{aligned}
\lim_{t \to 0^+} \frac{1 - F_{\text{radial}}(1 - u\,t\,)}{1 - F_{\text{radial}}(1 - t)} &= \lim_{t \to 0^+} \frac{1 - (1 - K(ut)^{\frac{d-1}{2}}) + o((t)^{\frac{d-1}{2}})}{1 - (1 - K(t)^{\frac{d-1}{2}}) + o((t)^{\frac{d-1}{2}})} \\
&= \lim_{t \to 0^+} \frac{K(ut)^{\frac{d-1}{2}} + o((t)^{\frac{d-1}{2}})}{K(t)^{\frac{d-1}{2}} + o((t)^{\frac{d-1}{2}})} \\
&= u^{\left(\frac{d-1}{2}\right)},
\end{aligned}
\tag{69}
$$

which guarantees convergence and in the same time gives the value of $\gamma = -\frac{2}{d-1}$.

To find suitable sequences $a_n$ and $b_n$, we can use the fact that $F_n(t) = F_{\text{radial}}(t)^n$ and study the behavior of $\ln F_n(t)$ near $t = 1$:

$$
\begin{aligned}
\ln F_n(t) &= \ln\left(F_{\text{radial}}(t)^n\right) \\
&= n \ln\left(F_{\text{radial}}(t)\right) \\
&= n(\ln\left(1 - K(1-t)^{\frac{-1}{\gamma}} + o((1-t)^{\frac{-1}{\gamma}})\right)) \text{ as } t \to 1^- \\
&= -nK((1-t)^{\frac{-1}{\gamma}} + o((1-t)^{\frac{-1}{\gamma}})) \text{ as } t \to 1^-.
\end{aligned}
\tag{70}
$$

By defining $a_n = -\gamma(Kn)^\gamma$, $b_n = 1 - (Kn)^\gamma$ and doing the change of variable $u = \frac{t - b_n}{a_n}$, we see that:

$$
\begin{aligned}
t &= a_n u + b_n \\
&= 1 - (1 + \gamma u)(Kn)^\gamma.
\end{aligned}
\tag{71}
$$

Since for every $u$, $\lim_{n \to +\infty}(1 + \gamma u)(Kn)^\gamma = 0$ (recall that $\gamma < 0$), the term $o((1 - x)^{\frac{-1}{\gamma}})$ as $x \to 1^-$ is equivalent to $o(\frac{1}{n})$ as $n \to +\infty$. this means that:

$$
\begin{aligned}
\ln\left(F_n(a_n u + b_n)\right) &= -nK\left(((1 + \gamma u)(Kn)^\gamma)^{\frac{-1}{\gamma}} + o\left((\frac{1}{n})\right)\right) \text{ as } n \to +\infty. \\
&= -(1 + \gamma u)^{\frac{-1}{\gamma}} + o(1) \text{ as } n \to +\infty.
\end{aligned}
\tag{72}
$$

We can now consider the exponential of the above expression to get our asymptotic maximum distribution:

$$
\lim_{n \to +\infty} F_n(a_n u + b_n) = e^{-(1 + \gamma u)^{\frac{-1}{\gamma}}},
\tag{73}
$$

which concludes the proof. $\qquad\square$

**Corollary C.6.** *With $\Gamma : z \mapsto \int_0^\infty t^{z-1} e^{-t}\, dt$ the Gamma function (Abramowitz & Stegun, 1948), we have:*

$$
\mathbb{E}_{\mathcal{X}_n \sim \mathcal{U}(\mathcal{S}^{d-1})}\left[\max_i \langle \tilde{V}, X_i \rangle\right] = 1 - \frac{\Gamma(\frac{d+1}{d-1})}{2}\left(\frac{(d-1)B(\frac{1}{2}, \frac{d-1}{2})}{n}\right)^{\frac{2}{d-1}} + o(\frac{1}{n^{\frac{2}{d-1}}}).
\tag{74}
$$

*Proof.* According to the Portmanteau theorem (Billingsley, 2013), Lemma C.5 is equivalent to:

$$
\frac{\max_i \langle \tilde{V}, X_i \rangle - b_n}{a_n} \xrightarrow{D} \text{GEV}(\gamma),
\tag{75}
$$

where $\text{GEV}(\gamma)$ is a generalized extreme value distribution with shape parameter $\gamma$ (Gnedenko, 1943). Recall that if a sequence $Z_1, Z_2, \ldots$ of random variables converges in distribution to a random variable $Z$, then for all bounded continuous

function $\phi$, $\lim\limits_{n\to+\infty} \mathbb{E}\left[\phi(Z_n)\right] = \mathbb{E}\left[\phi(Z)\right]$. Since $\frac{\max_i\langle\tilde{V}, X_i\rangle - b_n}{a_n}$ is bounded for every $n$, we can consider the identity function for $\phi$ and obtain:

$$\lim_{n\to+\infty} \mathbb{E}_{\mathcal{X}_n\sim\mathcal{U}(\mathcal{S}^{d-1})}\left[\frac{\max_i\langle\tilde{V}, X_i\rangle - b_n}{a_n}\right] = \mathbb{E}\left[\text{GEV}(\gamma)\right] = \frac{\Gamma(1-\gamma)-1}{\gamma}. \tag{76}$$

Replacing $\gamma$, $a_n$ and $b_n$ by their respective expressions, it implies that:

$$\lim_{n\to+\infty} \frac{\mathbb{E}_{\mathcal{X}_n\sim\mathcal{U}(\mathcal{S}^{d-1})}\left[\max_i\langle\tilde{V}, X_i\rangle\right] - 1 + (Kn)^{-\frac{2}{d-1}}}{(Kn)^{-\frac{2}{d-1}}} + \Gamma(\frac{d-1}{d-1}) - 1 = 0$$

$$\implies \lim_{n\to+\infty} \frac{\mathbb{E}_{\mathcal{X}_n\sim\mathcal{U}(\mathcal{S}^{d-1})}\left[\max_i\langle\tilde{V}, X_i\rangle\right] - 1 + K^{-\frac{2}{d-1}}\Gamma(\frac{d-1}{d-1})}{n^{-\frac{2}{d-1}}} = 0. \tag{77}$$

Since $K^{-\frac{2}{d-1}} = \frac{1}{2}\left(\frac{(d-1)\text{B}(\frac{1}{2},\frac{d-1}{2})}{n}\right)^{\frac{2}{d-1}}$, this is equivalent to writing:

$$\mathbb{E}_{\mathcal{X}_n\sim\mathcal{U}(\mathcal{S}^{d-1})}\left[\max_i\langle\tilde{V}, X_i\rangle\right] - 1 + \frac{1}{2}\left(\frac{(d-1)\text{B}(\frac{1}{2},\frac{d-1}{2})}{n}\right)^{\frac{2}{d-1}}\Gamma(\frac{d-1}{d-1}) = o(\frac{1}{n^{\frac{2}{d-1}}})$$

$$\Leftrightarrow \mathbb{E}_{\mathcal{X}_n\sim\mathcal{U}(\mathcal{S}^{d-1})}\left[\max_i\langle\tilde{V}, X_i\rangle\right] = 1 - \Gamma(\frac{d-1}{d-1})\frac{1}{2}\left(\frac{(d-1)\text{B}(\frac{1}{2},\frac{d-1}{2})}{n}\right)^{\frac{2}{d-1}} + o(\frac{1}{n^{\frac{2}{d-1}}}). \tag{78}$$

We have thus obtained Equation (74), concluding the proof of the corollary. $\qquad\square$

## C.2. Proof of Proposition 4.3

We now return to Proposition 4.3. In this section we consider the case of vMF-exp when $d > 2$ and $X_i$ embeddings are uniformly distributed on $\mathcal{S}^{d-1}$. Under those assumptions:

$$P_{\text{vMF-exp}}(a \mid n, d, V, \kappa) = \frac{f_{\text{vMF}}(A \mid V, \kappa)\mathcal{A}(\mathcal{S}^{d-1})}{n} + \mathcal{O}(\frac{1}{n^{1+\frac{2}{d-1}}}). \tag{79}$$

*Proof.* Similarly to the 2 dimensional case, the definition of $P_{\text{vMF-exp}}(a \mid n, d, V, \kappa)$ is:

$$P_{\text{vMF-exp}}(A \mid n, d, V, \kappa) = \mathbb{E}_{\mathcal{X}_n\sim\mathcal{U}(\mathcal{S}^{d-1})}\left[\mathbb{P}(\tilde{V} \in \mathcal{S}_{\text{Voronoï}}(A \mid \mathcal{X}_{n+1}) \mid \tilde{V} \sim \text{vMF}(V, \kappa))\right], \tag{80}$$

which can be written using the PDF of the vMF distribution:

$$P_{\text{vMF-exp}}(A \mid n, d, V, \kappa) = \mathbb{E}_{\mathcal{X}_n\sim\mathcal{U}(\mathcal{S}^{d-1})}\left[\int_{\tilde{V}\in\mathcal{S}_{\text{Voronoï}}(A\mid\mathcal{X}_{n+1})} f_{\text{vMF}}(\tilde{V} \mid V, \kappa)\,\mathrm{d}\tilde{V}\right]. \tag{81}$$

As done in the 2D case, we study the Taylor expansion of $f_{\text{vMF}}$ near $A$:

$$\begin{aligned}
\forall\tilde{V} \in \mathcal{S}_{\text{Voronoï}}(A \mid \mathcal{X}_{n+1}), f_{\text{vMF}}(\tilde{V} \mid \kappa, V) &= C_d(\kappa)e^{\kappa\langle V,\tilde{V}\rangle}\\
&= C_d(\kappa)e^{\kappa\langle V,A\rangle}e^{\kappa\langle V,\tilde{V}-A\rangle}\\
&= f_{\text{vMF}}(A \mid V, \kappa)\sum_{i=0}^{\infty}\frac{(\kappa\langle V,\tilde{V}-A\rangle)^i}{i!}\\
&= f_{\text{vMF}}(A \mid V, \kappa)(1 + \kappa\langle V,\tilde{V}-A\rangle + R_1(\tilde{V})).
\end{aligned} \tag{82}$$

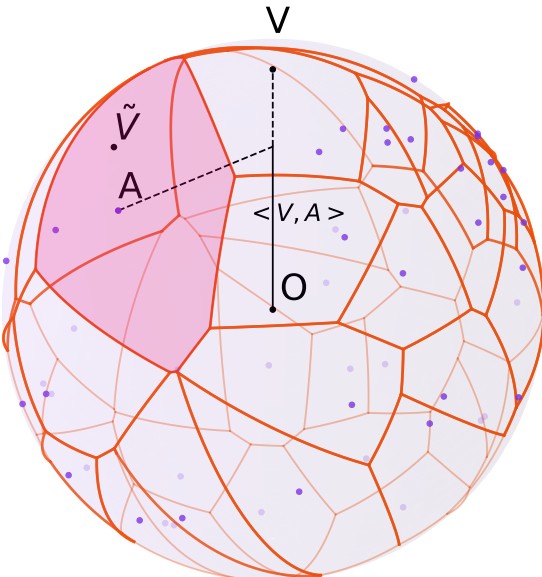

*Figure 6.* For $d = 3$: vMF-exp explores the action $A$ when $\tilde{V}$ lies in its Voronoï cell, shown in red.

with $R_1(\tilde{V}) = \sum_{i=2}^{\infty} \frac{(\kappa\langle V, \tilde{V} - A\rangle)^i}{i!}$. Leveraging the linearity property of both integration and expectation (Jacod & Protter, 2004), we can study $P_{\text{vMF-exp}}(A \mid n, d, V, \kappa)$ by assessing separately the contribution of the different terms of the expansion of $f_{\text{vMF}}$ in:

$$P_{\text{vMF-exp}}(A \mid n, d, V, \kappa) =$$
$$\mathbb{E}_{\mathcal{X}_n \sim \mathcal{U}(\mathcal{S}^{d-1})} \left[ \int_{\tilde{V} \in \mathcal{S}_{\text{Voronoï}}(A|\mathcal{X}_{n+1})} f_{\text{vMF}}(A \mid V, \kappa)(1 + \kappa\langle V, \tilde{V} - A\rangle + R_1(\tilde{V})) \, d\tilde{V} \right]. \tag{83}$$

However, contrary to the 2D case where $\mathcal{S}_{\text{Voronoï}}(A \mid \mathcal{X}_{n+1})$ is always defined as the arc between 2 angles on the circle, for $d > 2$ the shape of $\mathcal{S}_{\text{Voronoï}}(A \mid \mathcal{X}_{n+1})$ is highly dependent of the layout of the elements of $\mathcal{X}_n$ that share a frontier with $A$. Figure 6 provides an illustration of the complexity and diversity of the shapes of Voronoï cells for uniformly sampled points on the 3D sphere.

As a consequence, expliciting the bounds of integration, as we did in the 2D case, can be somewhat tedious. Instead, we will leverage the geometrical properties of the problem at hand to estimate $P_{\text{vMF-exp}}(A \mid n, d, V, \kappa)$. We start with the zero-order term.

### C.2.1. ZERO-ORDER TERM

Since the zero-order term is constant, its integral over $\mathcal{S}_{\text{Voronoï}}(A \mid \mathcal{X}_{n+1})$ can be expressed as:

$$\int_{\tilde{V} \in \mathcal{S}_{\text{Voronoï}}(A|\mathcal{X}_{n+1})} f_{\text{vMF}}(A \mid V, \kappa) \, d\tilde{V} = f_{\text{vMF}}(A \mid V, \kappa)\mathcal{A}(\mathcal{S}_{\text{Voronoï}}(A \mid \mathcal{X}_{n+1})), \tag{84}$$

where $\mathcal{A}(\mathcal{S}_{\text{Voronoï}}(A \mid \mathcal{X}_{n+1}))$ is the value of the surface area of $\mathcal{S}_{\text{Voronoï}}(A \mid \mathcal{X}_{n+1})$. To assess the expected value of the above equation for uniformly distributed $\mathcal{X}_n$, we use Lemma C.1 and obtain:

$$\mathbb{E}_{\mathcal{X}_n \sim \mathcal{U}(\mathcal{S}^{d-1})} \left[ \int_{\tilde{V} \in \mathcal{S}_{\text{Voronoï}}(A|\mathcal{X}_{n+1})} f_{\text{vMF}}(A \mid V, \kappa) \, d\tilde{V} \right] = \frac{f_{\text{vMF}}(A \mid V, \kappa)\mathcal{A}(\mathcal{S}^{d-1})}{n+1} \tag{85}$$

$$= \frac{f_{\text{vMF}}(A \mid V, \kappa)\mathcal{A}(\mathcal{S}^{d-1})}{n} + \mathcal{O}(\frac{1}{n^2}).$$

### C.2.2. FIRST-ORDER TERM

We want to estimate the value of:

$$\mathbb{E}_{\mathcal{X}_n \sim \mathcal{U}(\mathcal{S}^{d-1})} \left[ \int_{\tilde{V} \in \mathcal{S}_{\text{Voronoï}}(A|\mathcal{X}_{n+1})} f_{\text{vMF}}(A \mid V, \kappa) \kappa \langle V, \tilde{V} - A \rangle \, \mathrm{d}\tilde{V} \right] \tag{86}$$

$$= f_{\text{vMF}}(A \mid V, \kappa) \kappa \left( \langle V, \mathbb{E}_{\mathcal{X}_n \sim \mathcal{U}(\mathcal{S}^{d-1})} \left[ \int_{\tilde{V} \in \mathcal{S}_{\text{Voronoï}}(A|\mathcal{X}_{n+1})} \tilde{V} \, \mathrm{d}\tilde{V} \right] \rangle - \frac{\langle V, A \rangle \mathcal{A}(\mathcal{S}^{d-1})}{n} \right).$$

Using Lemmas C.2 and C.3 as well as Corollary C.6, the left term inside the parentheses is:

$$\langle V, \mathbb{E}_{\mathcal{X}_n \sim \mathcal{U}(\mathcal{S}^{d-1})} \left[ \int_{\tilde{V} \in \mathcal{S}_{\text{Voronoï}}(A|\mathcal{X}_{n+1})} \tilde{V} \, \mathrm{d}\tilde{V} \right] \rangle$$

$$= \langle V, A \rangle \frac{\mathcal{A}(\mathcal{S}^{d-1})}{n+1} \mathbb{E}_{\mathcal{X}_n \sim \mathcal{U}(\mathcal{S}^{d-1}), \tilde{V} \sim \mathcal{U}(\mathcal{S}^{d-1})} \left[ \max_i \langle \tilde{V}, X_i \rangle \right] \tag{87}$$

$$= \langle V, A \rangle \frac{\mathcal{A}(\mathcal{S}^{d-1})}{n+1} \left( 1 - \frac{\Gamma(\frac{d+1}{d-1})}{2} \left( \frac{(d-1)\mathrm{B}(\frac{1}{2}, \frac{d-1}{2})}{n} \right)^{\frac{2}{d-1}} + o(\frac{1}{n^{\frac{2}{d-1}}}) \right).$$

Re-injecting this expression into Equation (86) gives the following expression for the contribution of the first-order term to the probability of sampling $A$:

$$\mathbb{E}_{\mathcal{X}_n \sim \mathcal{U}(\mathcal{S}^{d-1})} \left[ \int_{\tilde{V} \in \mathcal{S}_{\text{Voronoï}}(A|\mathcal{X}_{n+1})} f_{\text{vMF}}(A \mid V, \kappa) \kappa \langle V, \tilde{V} - A \rangle \, \mathrm{d}\tilde{V} \right] \tag{88}$$

$$= -f_{\text{vMF}}(A \mid V, \kappa) \frac{\mathcal{A}(\mathcal{S}^{d-1})}{n+1} \kappa \langle V, A \rangle \left( \frac{\Gamma(\frac{d+1}{d-1})}{2} \left( \frac{(d-1)\mathrm{B}(\frac{1}{2}, \frac{d-1}{2})}{n} \right)^{\frac{2}{d-1}} + o(\frac{1}{n^{\frac{2}{d-1}}}) \right).$$

### C.2.3. REMAINDER TERM

As done in the 2D proof, we leverage the Taylor-Lagrange inequality (Abramowitz & Stegun, 1948). The second derivative of the function $f(x) = C_d(\kappa)e^{\kappa x}$ is $f(x)^{(2)} = \kappa^2 f(x)$, which is bounded on $x \in [-1, 1]$ by $M = \kappa^2 C_d(\kappa)e^{\kappa x}$. This implies that:

$$|R_1(\tilde{V})| \le \frac{M \langle V, \tilde{V} - A \rangle^2}{2}$$

$$\le \frac{M \|\tilde{V} - A\|_2^2}{2} \text{ (according to the Cauchy-Schwarz inequality (Jacod \& Protter, 2004))} \tag{89}$$

$$= M(1 - \langle \tilde{V}, A \rangle).$$

This inequality holds for every $\tilde{V} \in \mathcal{S}_{\text{Voronoï}}(A \mid \mathcal{X}_{n+1})$ when $\mathcal{X}_n \sim \mathcal{U}(\mathcal{S}^{d-1})$, which means that:

$$\mathbb{E}_{\mathcal{X}_n \sim \mathcal{U}(\mathcal{S}^{d-1})} \left[ \int_{\tilde{V} \in \mathcal{S}_{\text{Voronoï}}(A|\mathcal{X}_{n+1})} f_{\text{vMF}}(A \mid V, \kappa) |R_1(\tilde{V})| \, \mathrm{d}\tilde{V} \right]$$

$$\le f_{\text{vMF}}(A \mid V, \kappa) \mathbb{E}_{\mathcal{X}_n \sim \mathcal{U}(\mathcal{S}^{d-1})} \left[ \int_{\tilde{V} \in \mathcal{S}_{\text{Voronoï}}(A|\mathcal{X}_{n+1})} M(1 - \langle \tilde{V}, A \rangle) \, \mathrm{d}\tilde{V} \right]$$

$$= f_{\text{vMF}}(A \mid V, \kappa) \frac{\mathcal{A}(\mathcal{S}^{d-1})}{n+1} M(1 - \mathbb{E}_{\mathcal{X}_{n+1} \sim \mathcal{U}(\mathcal{S}^{d-1})} \left[ \int_{\tilde{V} \in \mathcal{S}^{d-1}} \max_i \langle \tilde{V}, X_i \rangle \, \mathrm{d}\tilde{V} \right]) \tag{90}$$

$$= f_{\text{vMF}}(A \mid V, \kappa) M \frac{\mathcal{A}(\mathcal{S}^{d-1})}{n+1} \left( \frac{\Gamma(\frac{d+1}{d-1})}{2} \left( \frac{(d-1)\mathrm{B}(\frac{1}{2}, \frac{d-1}{2})}{n} \right)^{\frac{2}{d-1}} + o(\frac{1}{n^{\frac{2}{d-1}}}) \right)$$

$$= O(\frac{1}{n^{1 + \frac{2}{d-1}}}).$$

We used Lemmas C.2 and C.3 to go from line 2 to 3, and Corollary C.6 to go from line 3 to 4. In essence, we have bounded the contribution of $R_1(\tilde{V})$ to the probability of sampling $A$ as follows:

$$\mathbb{E}_{\mathcal{X}_n \sim \mathcal{U}(\mathcal{S}^{d-1})} \left[ \int_{\tilde{V} \in \mathcal{S}_{\text{Voronoï}}(A|\mathcal{X}_{n+1})} f_{\text{vMF}}(A \mid V, \kappa) |R_1(\tilde{V})| \, \mathrm{d}\tilde{V} \right] = O(\frac{1}{n^{1 + \frac{2}{d-1}}}) \tag{91}$$

Finally, adding up Equations (85), (89), and (91), we conclude the proof of Proposition 4.3 for $d \geq 3$ and (via the first-order term) simultaneously justify the approximate probability $P_1(a \mid n, V, \kappa)$ introduced in Proposition 4.4. $\qquad\square$

# D. Similar Asymptotic Behavior of B-exp and vMF-exp for Large Action Sets (Proof of Proposition 4.1)

Finally, Propositions 4.2 and 4.3 allow us to derive Proposition 4.1, i.e., that in the setting of Section 4.1, we have:

$$\lim_{n \to +\infty} \frac{P_{\text{B-exp}}(a \mid n, d, V, \kappa)}{P_{\text{vMF-exp}}(a \mid n, d, V, \kappa)} = 1. \tag{92}$$

*Proof.* Acording to Proposition 4.2, we have:

$$P_{\text{B-exp}}(a \mid n, d, V, \kappa) = \frac{f_{\text{vMF}}(A \mid V, \kappa)\mathcal{A}(\mathcal{S}^{d-1})}{n} + o(\frac{1}{n\sqrt{n}}). \tag{93}$$

Moreover, according to Proposition 4.3, we have:

$$P_{\text{vMF-exp}}(a \mid n, d, V, \kappa) = \frac{f_{\text{vMF}}(A \mid V, \kappa)\mathcal{A}(\mathcal{S}^{d-1})}{n} + \begin{cases} \mathcal{O}(\frac{1}{n^2}) & \text{if } d = 2, \\ \mathcal{O}(\frac{1}{n^{1+\frac{2}{d-1}}}) & \text{if } d > 2. \end{cases} \tag{94}$$

Therefore:

$$\begin{aligned}
\lim_{n \to +\infty} \frac{P_{\text{B-exp}}(a \mid n, d, V, \kappa)}{P_{\text{vMF-exp}}(a \mid n, d, V, \kappa)} &= \lim_{n \to +\infty} \frac{n}{n} \frac{P_{\text{B-exp}}(a \mid n, d, V, \kappa)}{P_{\text{vMF-exp}}(a \mid n, d, V, \kappa)} \\
&= \frac{f_{\text{vMF}}(A \mid V, \kappa)\mathcal{A}(\mathcal{S}^{d-1}) + 0}{f_{\text{vMF}}(A \mid V, \kappa)\mathcal{A}(\mathcal{S}^{d-1}) + 0} \\
&= 1.
\end{aligned} \tag{95}$$

$\square$

# E. Link with Thompson Sampling

At first glance, one might draw some similarities between vMF-exp and Thompson Sampling (TS) with Gaussian prior for contextual bandits (Chapelle & Li, 2011). Admittedly, vMF-exp shares a common spirit with TS, where action selection is preceded by sampling individual weights according to a Normal distribution centered on an observed context/state vector. However, vMF-exp also presents two major differences:

- Firstly, in vMF-exp, vector sampling is performed according to a vMF hyperspherical distribution, centered on the state embedding vector $V$. This choice of distribution ensures that vectors with the same inner product with the state vector have the same probability of being sampled, as illustrated in Figure 1(a). This aligns better with the similarity used to retrieve nearest neighbors and, as emphasized in this paper, leads to probabilities of exploring actions asymptotically comparable to Boltzmann Exploration (with better scalability) under the theoretical assumptions of Section 4.1.

- Secondly, vMF-exp is not designed to maximize the expected reward of a policy in an RL or contextual bandit environment and does not impose any parameter update strategy. Instead, it serves as an action selection tool for any scenario where policy updates cannot be performed regularly (as in the batch RL setting commonly found in industrial applications), yet broad exploration must still be guaranteed between consecutive updates.

# F. Sampling from the von Mises-Fisher Distribution

### F.1. Radial-tangent decomposition

Given a vector $\tilde{V} \in \mathcal{S}^{d-1}$ and a concentration $\kappa \in \mathbb{R}^+$, the algorithm described in (Pinzón & Jung, 2023) sample from a vMF$(V, \kappa)$ by leveraging the radial-tangent decomposition of the elements of $\mathcal{S}^{d-1}$. For any $\tilde{V} \in \mathcal{S}^{d-1}$, let us call $\tilde{t} = \langle V, \tilde{V} \rangle$. Then we have:

$$\tilde{V} = \tilde{t}V + \sqrt{1 - \tilde{t}^2}\tilde{V}_O \tag{96}$$

where the vector $\tilde{V}_O$ has a unit norm and is orthogonal to $V$.

### F.2. vMF distribution

If, $\tilde{V} \sim$ vMF$(V, \kappa)$, then :

- $\tilde{t}$ is a scalar valued random variable.

- $\tilde{V}_O$ is a random vector uniformly distributed on the (d-2) dimensional sub-sphere that is centered at and perpendicular to $V$. For instance, for $d = 3$, this would mean a circle centered around $V$.

- $\tilde{t}$ and $\tilde{V}_O$ are independent.

Since the reciprocal is also true, $t$ and $\tilde{V}_O$ can thus be separately sampled to obtain $\tilde{V}$.

### F.3. Sampling $\tilde{t} = \langle V, \tilde{V} \rangle$

The PDF of $\tilde{t}$ is known (Fisher, 1953) and follows:

$$f_{\text{radial}}(t; \kappa, d) = \frac{(\kappa/2)^{\frac{d}{2}-1}}{\Gamma(\frac{1}{2})\Gamma(\frac{d-1}{2})I_{\frac{d}{2}-1}(\kappa)} e^{t\kappa}(1 - t^2)^{\frac{d-3}{2}} \tag{97}$$

This PDF can be used to sample r through rejection sampling (Gentle, 2009).

### F.4. Sampling $\tilde{V}_O$

$\tilde{V}_O$ can be obtained by following the steps of algorithm 1.

---

**Algorithm 1** Sample $\tilde{V}_O$

---

**1 -** Sample vector $U$ uniformly from $\mathcal{S}^{d-1}$;
**2 -** Compute projection of $U$ on $V$: $W = \langle U, V \rangle V$;
**3 -** Subtract projection and normalize: $\tilde{V}_O = \frac{U-W}{||U-W||}$;
**4 - return** $\tilde{V}_O$

---

Note that a simple way to sample U uniformly on $\mathcal{S}^{d-1}$ is to sample $d$ standard Gaussians independently (one for each dimension) and then normalize the resulting vector (Gentle, 2009).

### F.5. Wrapping up

The vector $\tilde{V}$ can now be computed by summing the right term of equation 97. Overall, we see that sampling $\tilde{V}$ from vMF$(V, \kappa)$ is **data-independent**, hence the scalability of the approach.

# G. Additional Monte Carlo Simulations

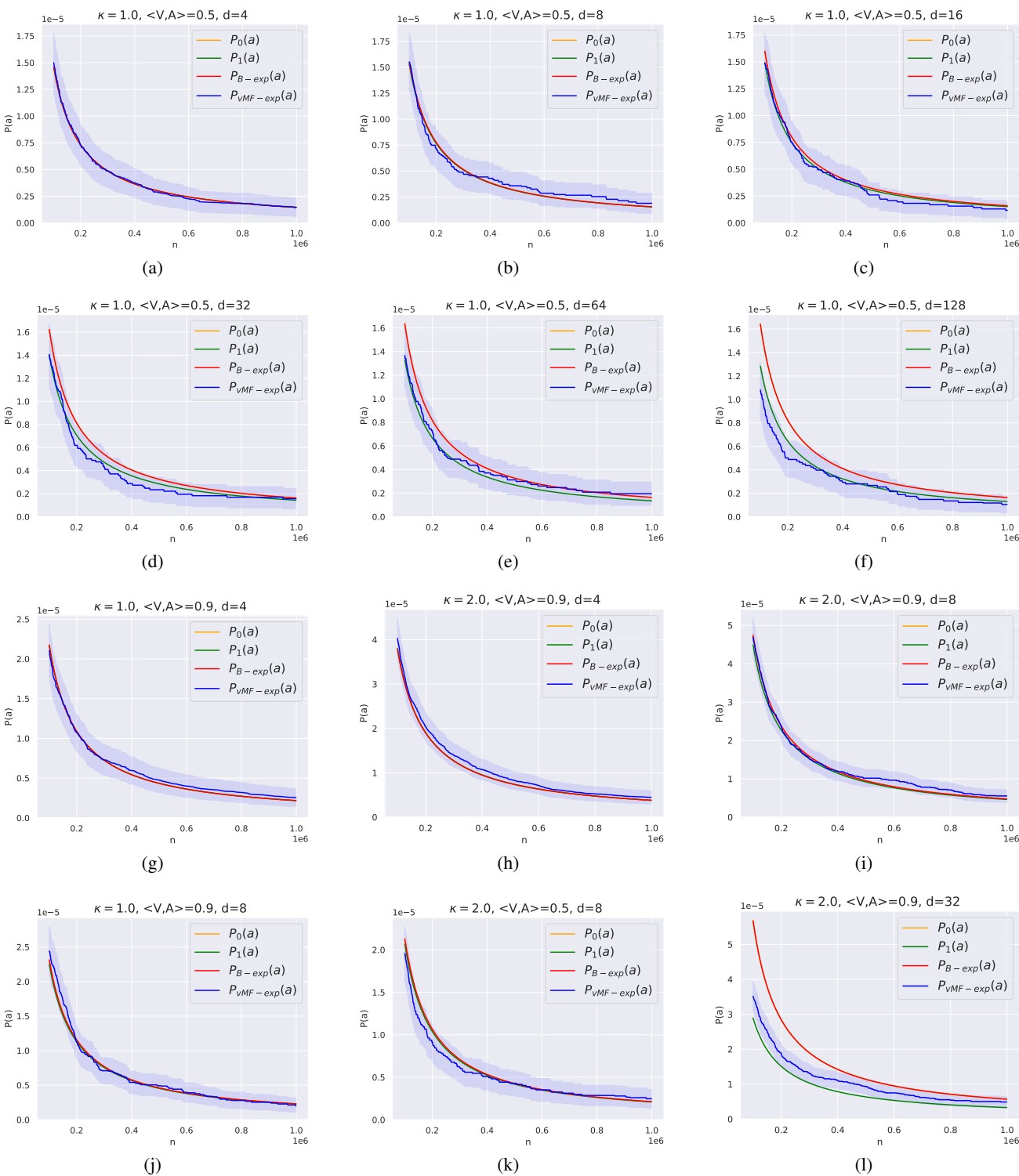

*Figure 7.* We report complete results for the Monte Carlo simulations presented and discussed in Section 4.3, involving more combinations of $d$, $\kappa$, and $\langle V, A \rangle$). We recall that $P_{\text{B-exp}}(a)$ and $P_0(a)$ are indistinguishable for this range of $n$ values. We emphasize that the y-axis is on a 1e-5 scale; hence, all probabilities are extremely close.

# H. Additional Experiments on a Real-World Dataset of GloVe Word Embedding Vectors

While our main contributions in this work are theoretical, we aimed in the main paper to validate our key findings with Monte Carlo simulations, which involved synthetic data. Understanding that some readers may wish to further explore our topic through reproducible experiments on real-world data, we present an additional study in this Appendix H. This study experimentally validates the main properties of vMF-exp on a large-scale, publicly available real-world dataset.

## H.1. Experimental setting

We present vMF-exp experiments on real-world, publicly available data. Specifically, we compare the behaviors of B-exp and vMF-exp on the GloVe-25 dataset of 1 million GloVe word embedding vectors with dimension $d = 25$ (Pennington et al., 2014). Each vector, learned using word2vec (Mikolov et al., 2013) from 2 billion tweets, represents a word token. We subtract the set's average from each vector and divide them by their norms. We obtain a vector set, denoted $\mathcal{G}$, with all vectors lying on the unit hypersphere, making GloVe-25 a relevant large-scale dataset for our study.

Our experiments follow the protocol outlined in Section 4.3 of the main paper. In this section, we compared the empirical probabilities $P_{\text{B-exp}}(a)$ and $P_{\text{vMF-exp}}(a)$ of sampling an action $a$ represented by a vector $A$ given a state vector $V$, for varying action numbers $n$ and inner products $\langle V, A \rangle$. While we relied on Monte Carlo simulations with uniformly drawn vectors $\mathcal{X}_n \sim \mathcal{U}(S^{d-1})$, in this Appendix H, vectors are sampled from $\mathcal{G}$, with $V$ and $A$ also drawn from $\mathcal{G}$ such that $\langle V, A \rangle$ matches the pre-selected values. Our goal is to empirically compare $P_{\text{B-exp}}(a)$ and $P_{\text{vMF-exp}}(a)$, while verifying the claims that **P1**, **P2**, and **P3** simultaneously hold for vMF-exp.

Finally, in our Sections 4.2 and 4.3, we provided analytical approximations of $P_{\text{B-exp}}(a)$ and $P_{\text{vMF-exp}}(a)$ in the presence of independent and identically distributed (i.i.d.) uniform embedding vectors. We will assess the usefulness of these approximations on these GloVe vectors, which do not strictly satisfy these strong assumptions.

Finally, regarding these experiments on GloVe-25, we note that:

- The GloVe-25 dataset is available for download at: `https://nlp.stanford.edu/projects/glove/`.

- In our experiments, we use the Python vMF sampler from Pinzón & Jung (2023) to efficiently explore large action sets.

- All results are reproducible using our source code: `https://github.com/deezer/vMF-exploration`.

## H.2. Results and Discussion

**On P1**    We now discuss our results. We first focus on **P1**. While B-exp requires computing $\langle V, X_i \rangle$ and softmax values for all $n$ vectors $X_i \in \mathcal{X}_n$, vMF-exp only involves sampling a $d$-dimensional vector (in constant time with respect to $n$) and finding its approximate nearest neighbor (ANN) in $\mathcal{X}_n$.

Table 1 compares the performance of four popular ANN algorithms on GloVe-25. Following standard ANN literature (Simhadri et al., 2024), our performance metric is the maximum throughput, measured in queries per second (QPS), for which the average recall of the exact top-10 neighbors exceeds 90%. We also report the throughput of exhaustive search, as an indicator of B-exp's inefficiency.

Table 1 shows that exhaustive search yields throughput 2 to 3 orders of magnitude lower than ANN methods. This confirms the significantly better scalability (**P1**) of vMF-exp compared to B-exp.

*Table 1.* Performance of popular ANN algorithms on GloVe-25, extracted from the benchmark of Aumüller et al. (2017). Following Simhadri et al. (2024), our performance metric is the maximum throughput, measured in Queries Per Second (QPS), for which the average recall of the exact top-10 neighbors exceeds 90%. The evaluated algorithms include two implementations of HNSW (Malkov & Yashunin, 2018), one from the Faiss library (Douze et al., 2024) and the other from NMSLIB (Boytsov & Naidan, 2013), as well as ScaNN (Guo et al., 2020) and NGT-QG (Iwasaki & Miyazaki, 2018). Exhaustive search is 2 to 3 orders of magnitude slower than ANN methods.

| Algorithm | Exhaustive Search | HNSW (Faiss) | HNSW (NMSLIB) | ScaNN | NGT-QG |
|---|---|---|---|---|---|
| **Maximum Throughput in Queries Per Second (QPS)** | 34 | 6197 | 14080 | **23436** | **22733** |

**On P2 and P3**   Figure 8 compares $P_{\text{B-exp}}(a)$ and $P_{\text{vMF-exp}}(a)$ for increasing values of $n$ and $\langle V, A \rangle$. Figure 8(a) highlights the two properties that make B-exp popular in RL: the ability to sample actions with unrestricted radius (**P2**) and the ordering of sampling probabilities based on action similarity to $V$ (**P3**).

Importantly, Figure 8(b) confirms that vMF-exp also satisfies both properties. In our tests, $A$ always has a positive sampling probability, which strictly increases with $\langle V, A \rangle$. Thus, vMF-exp also satisfies **P2** and **P3** on GloVe-25.

**On Theoretical Approximations**   Finally, Figure 9 shows that, although GloVe vectors are not i.i.d. and uniform, the analytical approximations of the main paper for $P_{\text{B-exp}}(a)$ and $P_{\text{vMF-exp}}(a)$ often remain accurate, particularly for B-exp. Also, vMF-exp closely matches B-exp's probabilities for low absolute values of $\langle V, A \rangle$.

However, as $|\langle V, A \rangle|$ increases, the gap between $P_{\text{vMF-exp}}(a)$ and $P_{\text{B-exp}}(a)$ grows more rapidly than predicted by approximations, highlighting the limitations of the i.i.d. and uniform assumptions and opening the way for future research on more general expressions.

**Conclusion**   This additional study confirmed the key theoretical and scalability properties of vMF-exp on a large-scale and publicly available real-world dataset. Our results highlight its potential as a practical solution for exploring large action sets when hyperspherical embedding vectors represent these actions.

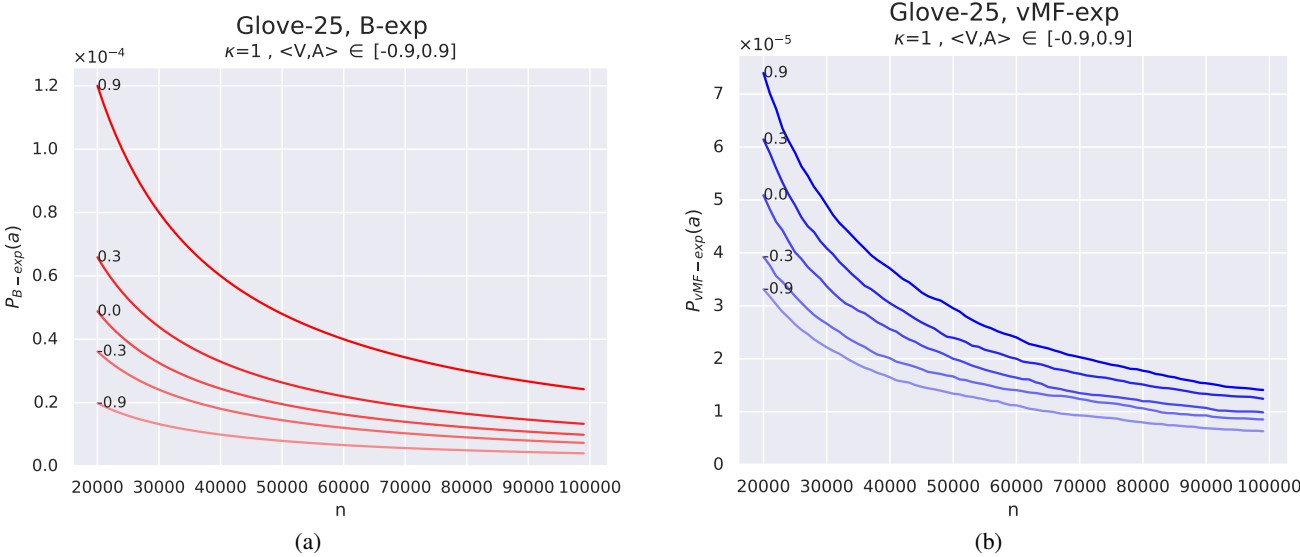

*Figure 8.* We report the empirical probabilities $P_{\text{B-exp}}(a)$ and $P_{\text{vMF-exp}}(a)$ of sampling an action $a$ represented by a GloVe embedding vector $A$, using B-exp and vMF-exp, respectively, given a state vector $V$, with $20000 \leq n \leq 100000$ and $\langle V, A \rangle \in \{0.9, 0.3, 0.0, -0.3, -0.9\}$, and with $d = 25$ and $\kappa = 1$. Sampling is repeated 30 million times and averaged to obtain precise estimates. For both methods, the probability of sampling $a$ for exploration is strictly positive (**P2**) and is a strictly increasing function of the inner product similarity $\langle V, A \rangle$ (**P3**). Results remain consistent when $\kappa$ is modified.

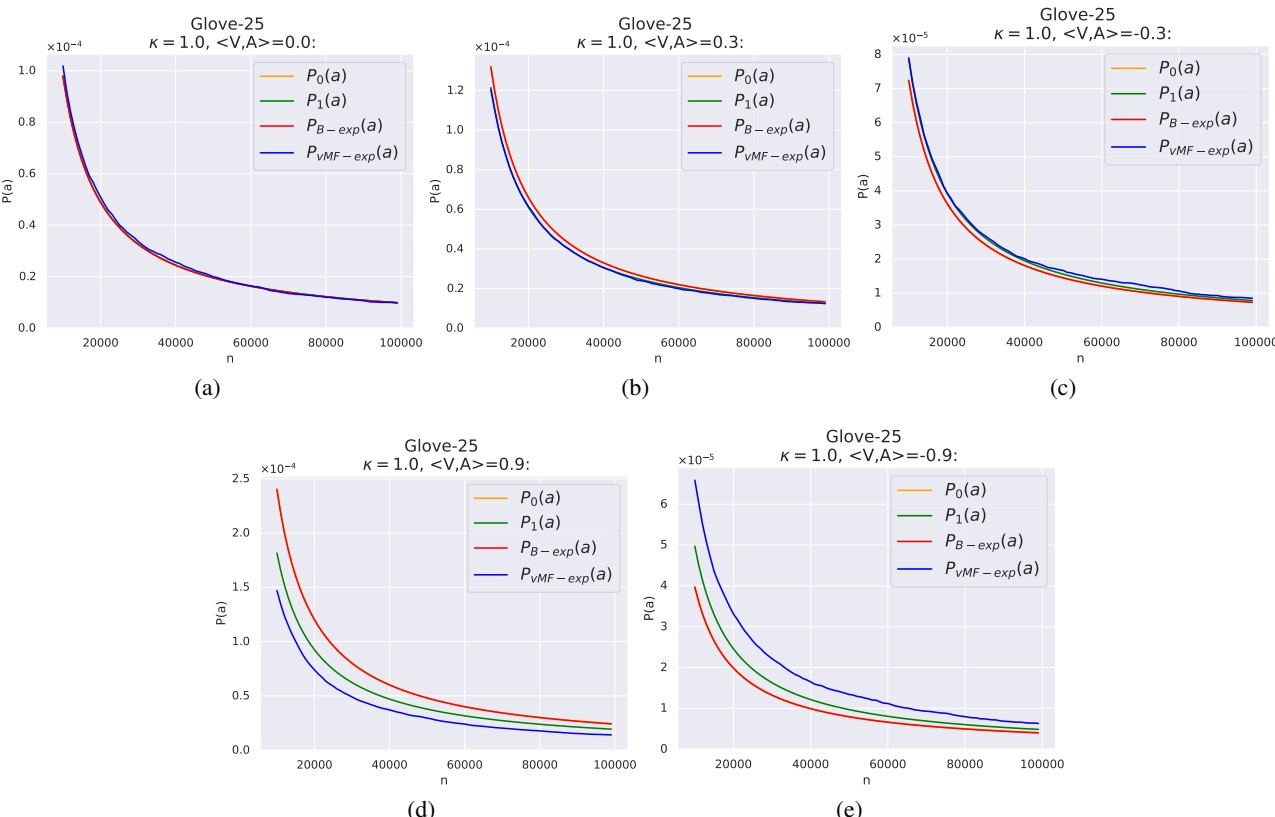

*Figure 9.* We compare $P_{\text{B-exp}}(a)$ and $P_{\text{vMF-exp}}(a)$ on GloVe-25 to the analytical approximations $P_0(a)$ and $P_1(a)$ stated in Propositions 4.1 to 4.4 of the main paper under the assumption of i.i.d. and uniformly distributed vectors. Our tests confirm the usefulness of these approximations on GloVe-25. The yellow curve ($P_0(a)$) is indistinguishable from the red curve ($P_{\text{B-exp}}(a)$), indicating that Proposition 4.2 holds across all configurations. Furthermore, $P_{\text{vMF-exp}}(a)$ (blue) remains close to $P_{\text{B-exp}}(a)$ (red) for low absolute values of $\langle V, A \rangle$ (Figures 9(a), 9(b), and 9(c)), as anticipated by Propositions 4.1 and 4.3. In Figures 9(b) and 9(c), the small difference between $P_{\text{vMF-exp}}(a)$ and $P_{\text{B-exp}}(a)$ aligns with the alternative expression $P_1(a)$ (green) derived in Proposition 4.4. However, as $|\langle V, A \rangle|$ increases (Figures 9(d) and 9(e)), the difference between vMF-exp and B-exp grows more rapidly than predicted by Proposition 4.4, highlighting the limitations of the i.i.d. and uniform assumptions.

# I. Application to Large-Scale Music Recommendation

Our analysis of vMF-exp in this paper was intentionally general, as the method can be applied to various problem settings. In this Appendix I, we showcase a real-world application of vMF-exp.

## I.1. Experimental Setting

We consider the "Mixes inspired by" feature of the global music streaming service Deezer[5]. This recommender system is deployed at scale and available on the homepage of this service (Bendada et al., 2023a). As shown in Figure 10, it displays a personalized shortlist of songs, selected from those previously liked by each user. A click on a song generates a playlist of 40 songs "inspired by" the initial one, with the aim of helping users discover new music within a catalog including several millions of recommendable songs.

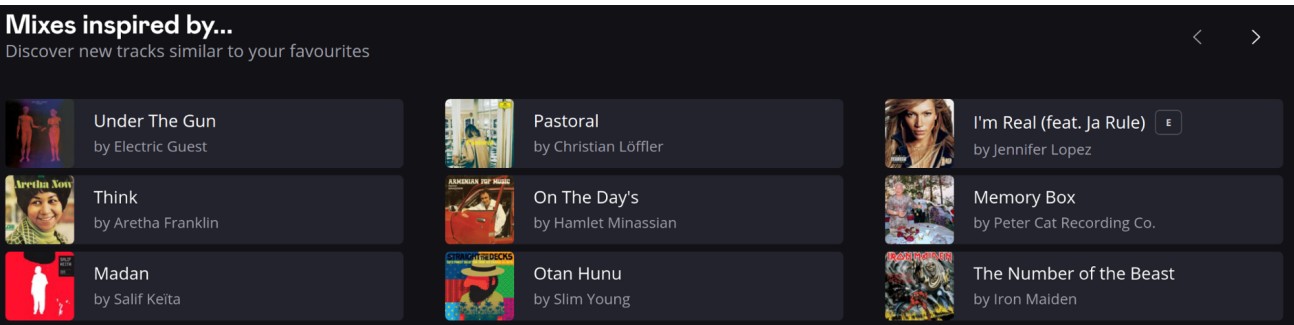

*Figure 10.* Interface of the "Mixes inspired by" recommender system on the music streaming service Deezer.

To generate playlists, Deezer leverages a collaborative filtering model (Koren & Bell, 2015). This model learns unit norm song embedding representations of dimension $d = 128$ by factorizing a mutual information matrix based on song co-occurrences in various listening contexts, using singular value decomposition (SVD) (Banerjee & Roy, 2014; Briand et al., 2021; 2024). Inner product proximity in the resulting embedding space aims to reflect user preferences. When a user selects an initial song, the model retrieves its embedding, then (approximately) identifies its neighbors in the embedding space using the efficient Faiss library (Johnson et al., 2019) for ANN. Currently, Deezer generates the entire playlist at once in production.

The service is considering RL approaches to, instead, recommend songs one by one while adapting to user feedback on previous songs of the playlist (likes, skips, etc.). However, as explained in Section 1, adopting such approaches would require exploring millions of possible actions/songs, significantly increasing the complexity of this task.

In this Appendix I, we continue generating "Mixes inspired by" playlists all at once, but take a step towards RL by comparing three methods for exploring large action sets of millions of songs:

- vMF-exp: we use the embedding of the user's selected song as the initial state $V$. We sample a random state embedding $\tilde{V}$ according to the vMF distribution, using the estimator of Banerjee et al. (2005) to tune $\kappa$ (see Equation (4) of Sra (2012)). Finally, we recommend the 40 nearest neighbors of $\tilde{V}$ in the embedding space according to the ANN engine.

- TB-exp: comparing vMF-exp to full B-exp is practically intractable at this scale. We compare vMF-exp to TB-exp with a similar $\kappa$. We first retrieve the $m = 500$ nearest neighbors of the initial song in the embedding space, according to the ANN engine. Then, we generate the playlist by sampling 40 songs from these 500 using a truncated Boltzmann distribution.

- Reference: we also compare vMF-exp to a baseline that retrieves the 500 nearest neighbors of the initial song using ANN, then shuffles them randomly to generate a playlist of 40 songs.

In early 2024, we conducted an industrial-scale online A/B test on the music streaming service Deezer to compare these exploration strategies in real conditions. The test involved millions of users worldwide, randomly split and unaware of the test.

---

[5]https://www.deezer.com/en/

## I.2. Results and Discussion

Firstly, it is important to highlight that we were able to successfully deploy vMF-exp in Deezer's production environment, achieving a sampling latency of just a few milliseconds, comparable to the other methods. This industrial deployment on a service used by millions of users on a daily basis confirms the claimed scalability of vMF-exp and its practical relevance for large-scale applications.

Using vMF-exp or TB-exp for exploration improved the daily number of recommended songs "liked" by users through "Mixes inspired by" (liking a song adds it to their list of favorites), compared to the reference baseline. For confidentiality, we do not report exact numbers of likes or users in each cohort, but present relative rates with respect to the reference. On average, users exposed to vMF-exp or TB-exp added 11% more recommended songs to their playlists than the reference cohort. These differences were statistically significant at the 1% level (p-value $< 0.01$). No apparent differences were observed between vMF-exp and TB-exp, showing that vMF-exp is competitive with TB-exp.

In addition, vMF-exp, which does not suffer from the restricted radius of TB-exp, recommended more diverse playlists. We measured the average Jaccard similarity (Tan et al., 2016) of playlists generated from the same initial selection, to assess how similar the songs sampled from the same state embedding were, for each method. Results reveal that TB-exp had an average Jaccard similarity 35% higher (less diverse playlists) than vMF-exp, a statistically significant difference at the 1% level (p-value $< 0.01$). Therefore, vMF-exp allowed for a more substantial exploration, without compromising performance.

At the time of writing, Deezer continues to use vMF-exp for "Mixes inspired by" recommendations. Playlists are still generated at once, but our work equips this service with an effective strategy to explore their large and embedded action set of millions of songs. This opens interesting avenues for further investigation of RL for recommendation. In the near future, Deezer will launch tests involving actor-critic RL models (Konda & Tsitsiklis, 1999; Sutton & Barto, 2018) to explore and generate songs sequentially based on user feedback.

