# OpenReview forum: "Exploring Large Action Sets with Hyperspherical Embeddings using von Mises-Fisher Sampling"
_ICML.cc/2025/Conference — ICML 2025 poster_

### Official Review · Reviewer_Rb4B · 2025-03-12

**Overall Recommendation:** 3

**Summary:**

The paper considers exploration in large action spaces where simple baselines like epsilon-greedy are impractical. The authors motivate that prior SoTA on this problem uses approximate nearest neighbor search to inform a truncated version of Boltzmann exploration, which does not have a clean theoretical characterization. By contrast, the authors propose Von Mises Fisher  exploration to directly sample the action embedded in a unit sphere, with exploration asymptotically matching that of Boltzmann exploration. This is combined with experimental evidence to support the theory along with a theoretical comparison to boltzmann exploration.

**Claims And Evidence:**

Yes.

**Essential References Not Discussed:**

n/a

**Ethical Review Flag:**

Flag this paper for an ethics review.

**Experimental Designs Or Analyses:**

The authors explore the performance for non iid embeddings like the Glove Vectors in the appendix and a real world deployment where they claim it showed benefits, though the baseline is unclear. While the authors compare with Boltzmann exploration, given the motivation in the introduction it would be interesting to also compare to the truncated boltzmann distribution (similar to the wolpertinger policy) which seems to be missing.

**Methods And Evaluation Criteria:**

Yes

**Other Comments Or Suggestions:**

n/a

**Other Strengths And Weaknesses:**

Strengths: Simple exploration strategy that's intuitively easy to appreciate and simple to implement. Proof of evidence of deployment with clear benefits recognized in a production setting is impressive.

Weaknesses: The justification for why this is a good idea makes a uniformity assumption on the embeddings which doesn't seem very accurate, and more importantly, the proposed reasoning for why this helps does not appear to hold when violating this assumption. It is plausible that the scheme is still a good idea for other reasons, but I am not sure the current theory indicates anything of that sort.

**Questions For Authors:**

Have you considered interpreting the algorithm as accounting for uncertainty in the query embedding?  I wonder if that might give an alternate potential justification for why this could be a good idea, especially when the catalogue embeddings are not uniformly distributed on the hyper-sphere.

**Relation To Broader Scientific Literature:**

The authors introduce a new and simple exploration strategy that involves sampling a unit embedding from the vonmises fisher distribution and then running a cheap ann search as the exploration mechanism. This is in contrast to prior work that proposed truncated Boltzmann exploration for similar problems (TB-exp with several references listed in the paper)

**Theoretical Claims:**

I believe the assumption of uniformity in the distribution of the action embeddings makes the result less interesting. On the other hand, with a heavily imbalanced sample of the action embeddings, a very low probability embedding could nevertheless end up being sampled much more significantly with the proposed strategy due to a large voronoi cell attracting the nearest neighbor search to it, which would nevertheless not get sampled in vanilla Boltzmann exploration.

Having said that, I believe the authors allude to something like this in the limitations and future work section.

---

> ### Author Rebuttal · Authors · 2025-03-31
>
> **On the interpretation of the algorithm**
>
> The interpretation of our algorithm as a way of accounting for the uncertainty in the query embedding is extremely interesting and will be further investigated in subsequent work. Indeed, the operation of sampling $\tilde{V}$ from a directional distribution, such as von Mises-Fisher, around the initial state vector $V$ can be understood as the embedding equivalent of reformulating a query in information retrieval tasks. With our method, the extent to which the query embedding can be reformulated is entirely controlled by the parameter $\kappa$, which can either remain constant across the hypersphere (as in our offline experiments in Section 4.3 and Appendix H) or depend on additional information, such as the local density of embeddings (as in the real-world A/B test detailed in Appendix I).
>
> As the need for reformulation generally arises when the intent behind the query is ambiguous, this could explain why our method benefits recommender systems based on vector searches. Since users do not explicitly formulate their needs, the system must infer them from past user actions, leaving plenty of room for uncertainty. To investigate this hypothesis, in future experiments, we will consider calibrating the concentration parameter $\kappa$ of the von Mises-Fisher distribution to an independent estimate of the uncertainty of the current query embedding. The impact of such an approach on the observed reward should provide valuable insights into the relevance of this interpretation.
>
>
> **On the uniform distribution assumption**
>
> This aspect was also noted by Reviewer sVYv, and overall, we agree that the assumption of a uniform distribution over the hypersphere is strong and may not hold in practical settings. However, we emphasize that the vMF-exp method itself remains fully applicable regardless of the distribution of action embeddings. Most of its key properties, such as scalability and unrestricted radius, do not rely on the uniformity assumption. Uniformity is used solely to derive the asymptotic equivalence with B-exp and to obtain the analytical approximations presented in Propositions 4.1 to 4.4. Furthermore, the non-trivial proof detailed from Appendices A to D provides a useful foundation for future attempts at deriving a more general result akin to Propositions 4.1 to 4.4 under relaxed assumptions.
>
> Interestingly, our experiments on the publicly available GloVe word embedding dataset (Appendix H) suggest that, although these real-world vectors are clearly not uniformly distributed, the analytical approximations from Propositions 4.1 and 4.4 remain accurate in most cases. Moreover, the A/B testing conducted on the music streaming service (Appendix I) further showcases the practical effectiveness of vMF-exp in an industrial-scale, real-world scenario where, again, the action embeddings are not sampled uniformly.
>
> Taken together, these results highlight the robustness and practical relevance of vMF-exp beyond the idealized uniform setting of our theoretical study.
>
> **On the case of heavily imbalanced samples**
>
> It is true that, under a heavily imbalanced distribution of action embeddings, actions associated with larger Voronoi cells may "attract the nearest neighbor search to them." However, we emphasize that the sampling density of the vMF distribution is still modulated by the inner product similarity between the state and action embeddings. In other words, even if an action has a large Voronoi cell, a very low similarity with the current state will significantly reduce its sampling probability. This mitigates the risk of over-sampling irrelevant actions solely due to geometric imbalance.
>
> Regarding how much more frequently such an action might be sampled compared to vanilla Boltzmann exploration, the answer is inherently distribution-dependent. The reviewer’s comment highlights an interesting direction for future theoretical investigation into the different behaviors of B-exp and vMF-exp under heavily imbalanced embedding distributions. We will consider discussing this aspect explicitly in the limitations and future work section of the paper.
>
> The outline of this future research agenda should be as follows:
>
> - Assess the extent to which real-world distributions exhibit outliers with large Voronoi cells.
> - Evaluate the impact of over-sampling isolated vectors (and under-sampling potentially redundant actions) on the training of models in a batch-learning setting.
> - Consider alternative distributions from directional statistics that allow for anisotropic sampling, such as the Fisher-Bingham distribution, to balance the oversampling of isolated points.
>
> More details about this research agenda can be found in our response to Reviewer sVYv.

---

### Official Review · Reviewer_xRgZ · 2025-03-12

**Overall Recommendation:** 4

**Summary:**

Summary: The authors propose to improve exploration for very large actions spaces (e.g., millions of samples). This work attempts to overcome the main limitation of Boltzmann exploration for high dimensional actions spaces which requires calculating cosine similarities between the reference sample and all other samples. The proposed exploration method using von Mises-Fisher sampling improves upon this by setting probabilities for exploration depending on whether a candidate sample is near the current sample according to a voronoi cell in a hypersphere. Thus the calculation for the exploration probabilities can be kept constant with regards to the number of actions.


### Update after rebuttal

I appreciate the authors responses. I think this is a very well written paper with clear desiderata that are needed for sampling in very large action spaces. Even though the experimental section is limited in its thoroughness, I think the authors rather convincingly show that their method is effective. I will keep my score of accept.

**Claims And Evidence:**

The paper is extremely well written. The authors propose a set of conditions that need to be met for efficient exploration in high-dimensional spaces and explain how their method fulfills all these conditions in comparison to other methods. Furthermore, an exhaustive analysis is performed of the exploration behaviour when using von Mises-Fisher sampling.

**Essential References Not Discussed:**

None

**Experimental Designs Or Analyses:**

I think the authors do a commendable effort on studying the effectiveness of their method and on whether it improves upon Boltzmann exploration. While concrete benchmarks results are not available, given the theoretical guarantees and the synthetic analysis it is likely that the proposed method will perform better.

The authors also provide a meta analysis of a field study performed on a large streaming service.

**Methods And Evaluation Criteria:**

The authors clearly identify issues with current Boltzmann exploration, which is its inefficiency when dealing with millions of samples. However, the authors go further by proposing a set of desiderate that an exploration algorithm should have for their intended purpose: Scalability, Unrestricted radius, Order preservation. I believe these criteria are sensible for the problem contstraints.

The authors scrutinise random, boltzmann and von mises-fisher exploration under these criteria and show that their proposed method fulfills all of these criteria.

**Other Comments Or Suggestions:**

None

**Other Strengths And Weaknesses:**

None

**Questions For Authors:**

- How well would the method work for embeddings composed of discrete features? As I understand it, von Mises-Fisher sampling relies on neighbourhoods between samples to be reasonably close to each other. What happens when features are distant in euclidean space.

**Relation To Broader Scientific Literature:**

I think this is an interesting problem to address from a technical perspective. However, the problem is quite constrained to the proposed issue of music recommendations. I wonder how well the von Mises-Fisher sampling would work when embeddings are not as "close" in euclidean space.

I would like to see the authors address more limitations regarding the type of embeddings that can be used with their proposed methods.

**Theoretical Claims:**

I did not check every claim in detail, although most justifications follow logically  if one is familiar with the discussed methods. In any case, the authors show in Figure 2, that their proposed method holds the properties that were derived for their proposed method.

---

> ### Author Rebuttal · Authors · 2025-03-31
>
> **On the possible distribution of embeddings**
>
> Regarding your concern, vMF sampling is applicable to a wide range of embedding distributions and is not restricted to the uniform distribution on the sphere, which we assumed in part of the theoretical analysis in Section 4. vMF sampling operates by identifying approximate nearest neighbors based on inner products in a d-dimensional hypersphere. Equivalently, it can be understood as finding nearest neighbors using cosine similarity in a d-dimensional Euclidean space. Unlike Euclidean distance, cosine similarity is inherently bounded between -1 and 1, making our approach robust even when embeddings are far apart in Euclidean terms.
> To assess the behavior of our method for both low and high inner product similarity values, we provide in Figure 9 (Appendix H) an analysis of vMF sampling on the real-world dataset of GloVe-25 embeddings, considering cases where the action to be sampled has a similarity of -0.9 and +0.9 with the context vector. These plots demonstrate that while vMF exploration behaves differently from Boltzmann exploration, it still satisfies properties P1, P2, and P3 outlined in Section 2, reinforcing its practical applicability.
>
>
> **On the particular case of discrete features**
>
> You also raise an interesting question regarding the behavior of vMF sampling when features are discrete. This is particularly relevant in the context of embedding quantization, which reduces memory usage and accelerates similarity computations by decreasing the number of bitwise comparisons. In this scenario, it is again important to highlight that cosine similarities remain bounded between -1 and 1, ensuring that approximate nearest neighbor search remains feasible.
> A potential challenge in highly quantized embeddings is that the set of possible inner product similarities becomes discrete, potentially leading to ambiguity when multiple points have identical similarities to the context vector. However, this issue arises even in Boltzmann exploration, as it also depends on computing inner product similarities before sampling actions. Moreover, even when embeddings are discrete, sampling from a vMF distribution around a discrete state embedding $V$ produces a new state vector $\tilde{V}$ with continuous features, mitigating the aforementioned issue.
>
>
> **On additional applications beyond music recommendation**
>
> It is worth noting that although the submission has recurrently referred to the example of music recommendation, other recommendation scenarios involving very large catalogs can be considered. To illustrate this, we refer to the paper [1] mentioned by Reviewer gBBS. In this paper, which deals with the task of extreme classification, a dataset of Amazon products with 670,000 different labels is considered for evaluation. More generally, extreme classification is an active area of research that involves different sorts of applications, and for which a benchmark can be found at http://manikvarma.org/downloads/XC/XMLRepository.html
>
> [1] Lopez et al., Learning from eXtreme Bandit Feedback, 2020

---

### Official Review · Reviewer_sVYv · 2025-03-26

**Overall Recommendation:** 3

**Summary:**

This paper addresses the challenge of exploration in reinforcement learning (RL) when the action space is extremely large, as in real-world recommendation systems like music streaming platforms. Traditional exploration strategies such as Boltzmann exploration and epsilon-greedy become inefficient or intractable in these settings, especially when millions of actions are involved and many are irrelevant. A common workaround, truncated Boltzmann exploration (TB-exp), limits exploration to a small subset of actions, but this may hinder optimal performance.

The authors propose a new method, von Mises-Fisher exploration (vMF-exp), designed for large-scale RL tasks where actions are represented by embedding vectors on a unit hypersphere. vMF-exp samples a direction using a von Mises-Fisher distribution and explores actions in that direction, allowing scalable and informative exploration. The paper provides a theoretical analysis showing that vMF-exp retains key properties of Boltzmann exploration in certain regimes while being computationally efficient. Empirical results, including real-world deployment on a music streaming platform, support the method’s effectiveness. A public Python implementation is also provided.

**Claims And Evidence:**

Claims and supported by mathematical statements and experiments.

**Essential References Not Discussed:**

NA

**Experimental Designs Or Analyses:**

I didn't check in detail.

**Methods And Evaluation Criteria:**

They seem to make sense.

**Other Comments Or Suggestions:**

Paper is well written and easy to follow.

**Other Strengths And Weaknesses:**

Main weakness: While vMF-exp offers a more scalable alternative to B-exp, its theoretical justification relies on a regime where action embeddings are sampled uniformly at random from the hypersphere—a setting that is unlikely to reflect the structure of real-world datasets. Moreover, vMF-exp tends to assign higher sampling probability to isolated actions (i.e., those with large Voronoi cells), which may not be desirable. For example, in a music recommendation context, this could lead to disproportionately sampling songs from niche genres unrelated to a user’s preferences. This behavior raises concerns about the method’s practical suitability, and as a result, I remain unconvinced that vMF-exp is the right scalable solution for exploration in large action spaces. With that said, experimental results still look promising.

**Questions For Authors:**

Can you address the weakness that I pointed out.

**Relation To Broader Scientific Literature:**

The works builds on the broader research effort of designing scalable exploration strategies in large action spaces.

**Theoretical Claims:**

I did not check the correctness of the proofs.

---

> ### Author Rebuttal · Authors · 2025-03-31
>
> **On the uniform distribution**
>
> The reviewer highlights an important point. We agree that the assumption of a uniform distribution over the hypersphere is strong and may not hold in practical settings. However, we would like to emphasize that the vMF-exp method itself remains fully applicable regardless of the distribution of action embeddings. Most of its key properties, such as scalability and unrestricted radius, do not rely on the uniformity assumption. Uniformity is used solely to derive the asymptotic equivalence with B-exp and to obtain the analytical approximations presented in Propositions 4.1 to 4.4.
>
> Interestingly, our experiments on the publicly available GloVe word embedding dataset (Appendix H) suggest that, although these real-world vectors are clearly not uniformly distributed, the analytical approximations from Propositions 4.1 and 4.4 remain accurate in most cases. Moreover, the A/B testing conducted on the music streaming service (Appendix I) further showcases the practical effectiveness of vMF-exp in an industrial-scale, real-world scenario where, again, the action embeddings are not sampled uniformly.
>
> Taken together, these results highlight the robustness and practical relevance of vMF-exp beyond the idealized uniform setting of our theoretical study.
>
>
> **On the potential over-sampling of isolated actions with large Voronoi cells**
>
> The reviewer raises an interesting question. In practice, a large Voronoi cell simply implies that a given action is more likely to be sampled than an equally undesirable action with a smaller Voronoi cell, but *not necessarily* that it would be sampled disproportionately overall. When Property P3 holds, the sampling probabilities remain aligned with action relevance. That is, undesirable actions, even those with large Voronoi cells, will still receive lower sampling probabilities than more desirable ones. Therefore, the ranking of actions by relevance is preserved, and the method does not favor undesirable actions in an uncontrolled manner.
>
> That said, the reviewer's comment opens up interesting avenues for additional research related to exploration in the
> presence of outliers. We detail below a proposal for a future research agenda tackling the specific problem of outliers management in the context of a the scalable sampling method proposed in this submission:
> * *Prevalence of isolated points in real-world datasets of embeddings:* the first step would be to determine if real-world embeddings actually exhibit isolated points with larger Voronoi cell overly attracting nearest neighbor search. Theoretical approaches using tools from statistical theory, such as Mardia's multivariate kurtosis, and empirical approaches based on Monte Carlo estimation of the surface of Voronoi cells can both be considered for this task.
> * *Impact of oversampling isolated actions on model training and reward*: in case that isolated points frequently occur, a follow-up research question would be to determine the impact on observed rewards and model training, which wouldn't necessarily be negative. Indeed, actions embedded close to one another are likely to provide a similar reward, thus carrying redundant information when selected. A method undersampling those actions in favor of isolated ones could thus provide more diverse batches for training, potentially accelerating convergence of bandit and reinforcement learning approaches, all the while introducing diversity in the selected actions which in the case of a recommender system can provide a more interesting experience.
> * *Correcting for oversampling of isolated points*: while the von Mises-Fisher distribution samples vectors isotropically around its mean direction, other distributions from the directional statistics literature can be considered. One interesting candidate could be the Fisher–Bingham distribution, whose probability density function is given by:
> $$f(\tilde{V} | {V}, {\Sigma}) = \frac{\exp ( - \frac{1}{2} \tilde{V}^\top {\Sigma} \tilde{V} + \tilde{V}^{\top}\Sigma V ) }{C({\mu}, {\Sigma} )} $$
> where $\Sigma$ is a full rank, square concentration matrix, that can be fitted from embeddings in the neighborhood of the context vector $V$ to account for anisotropy, thus favoring denser areas around $V$ containing more embeddings.

---

### Official Review · Reviewer_gBBS · 2025-03-26

**Overall Recommendation:** 3

**Summary:**

This paper proposes a method called vMF-exp: a method for exploration in tasks with large action spaces. One such task is recommender systems, where there are millions of categories to choose from. The paper discusses 3 important properties in order to have good exploration: 1. scalability to sample actions from a large search space; 2) high radius to have some probability of sampling every action; 3) order preservation where the probability of selection of an action depends on some similarity metric. The paper then discusses the problems with existing methods like $\epsilon$-greedy, Boltzmann exploration, and Truncated Boltzmann exploration~(TB-exp). A method called vMF-Exploration is proposed which satisfies these properties. A/B testing is conducted where the vMF-exp performs similarly to the TB-exp on a music recommendation platform.

**Claims And Evidence:**

The paper motivates the importance of efficient exploration for the application of recommender systems. However, the proposed method vMF performs similarly to the known truncated Boltzmann exploration. It is unclear is the proposed method helped with exploration in the action space.

It is unclear if the proposed method is helping with exploration in web-scale recommender systems. As discussed in the paper, TB-exp suffers from exploration issues because of the fixed radius while sampling exploratory action (P2), it would be interesting to see what fraction of actions were sampled outside this radius and how often that was a good action.

**Essential References Not Discussed:**

NA

**Experimental Designs Or Analyses:**

I checked Appendix I where the paper discusses the experiments and results of A/B testing.

**Methods And Evaluation Criteria:**

The paper does not compare with any open-source datasets used to test the claim over recommender systems. Another concern is that only A/B testing results are reported and there are no analysis on where the gains are coming from.

**Other Comments Or Suggestions:**

NA

**Other Strengths And Weaknesses:**

1. The paper is not well-written. The paper talks about importance of exploration in recommendation systems, but all the results of A/B testing are pushed to the Appendix. The main experiments should be in the main paper. The discussion on properties P1-P3 and the existing methods is verbose and having a simple table will help articulate this better.

2. Another concern is that the A/B testing results are not convincing. The paper would benefit from experiments on a public dataset.

3. How does vMF-exp compare to algorithms for eXtreme Multi-Label Classification (XC) like [1]?

[1] Lopez et al., Learning from eXtreme Bandit Feedback, 2020

**Questions For Authors:**

NA

**Relation To Broader Scientific Literature:**

Exploration is challenging in web-scale recommendation systems due to large action spaces. Moreover, A/B testing is hard as the random exploratory actions can be irrelevant. Any method that improves this exploration is applicable for retrieval in web-scale recommendation systems.

**Theoretical Claims:**

I checked Proposition 4.1-4.3.

---

> ### Author Rebuttal · Authors · 2025-03-31
>
> **Point 2**
>
> We begin with Point 2, which we understand to be the reviewer’s primary concern.
>
> This point appears to stem from a misunderstanding, as our paper does, in fact, include extensive experiments on a publicly available real-world dataset.
>
> Specifically, Section 5 and Appendix H present an in-depth experimental validation of the scalability and theoretical properties of vMF-exp, using a large-scale dataset of publicly available GloVe word embeddings. These experiments reinforce the conclusions of the paper. In particular, they demonstrate that actions consistently receive a positive sampling probability (Property P2 – unrestricted radius), which, by design, TB-exp fails to achieve. Moreover, the experiments validate the practical relevance of Properties 4.1 to 4.4 derived in our theoretical analysis when applied to real-world data.
>
> We have made our source code publicly available to ensure full reproducibility.
>
> We hope this clarification will help address the reviewer's reservations regarding the evaluation of the method.
>
> *P.S.* We plan to add a figure to the A/B test discussion to clearly illustrate that, in this A/B test setting as well, all actions receive a positive sampling probability under vMF-exp; unlike TB-exp, where probabilities are set to zero beyond the top-500 radius. We hope this additional plot will reinforce the message of the A/B test, which not only demonstrates the practical value of vMF-exp through its successful deployment at scale for music recommendation to millions of users, but also highlights its ability to support broader exploration without compromising performance.
>
> **Point 1**
>
> While two other reviewers found the paper to be *"extremely well written"* and *"well written and easy to follow,"*  we understand the reviewer's comment and would like to clarify the motivation behind the structure and focus of the paper, which may help contextualize our presentation choices.
>
> As stated in the paper, the primary objective of this work is to introduce vMF-exp and provide a rigorous mathematical analysis of its exploration behavior over large action sets. In this sense, the paper is positioned primarily as a theoretical contribution, one that could have been presented as a purely theoretical article. The experimental results were intended as a complement to the theoretical findings, offering empirical insights and illustrating the applicability of vMF-exp beyond mathematical results. This motivation guided our decision to place these experiments in the appendix in order to preserve focus in the main text.
>
> That said, we appreciate the reviewer's suggestion. We will consider creating a summary table to better articulate the key differences among methods and improve accessibility. Moreover, in a future arXiv version of the article, where space constraints are not an issue, we will consider moving both experiments on GloVe public data and the A/B test into the main body of the paper.
>
> **Point 3**
>
> The mentioned paper deals with correcting importance sampling estimators of bandit algorithms when the action set is extremely large. To reduce variance, they make the explicit assumption that most actions are irrelevant, assigning them a non-learnable minimal reward and stabilizing the computation of the expected reward for a learnable policy.
>
> However, they explicitly state that they do not provide a solution to identify the subset of relevant actions to explore, and instead resort to "a greedy heuristic," namely top-$p$ sampling with $p = 20$, followed by softmax, i.e., TB-exp with $m = 20$. Our exploration method could be combined with their training approach for a potentially stronger algorithm (in our case, the set of relevant actions can vary depending on $x$, which constitutes a meaningful improvement over the fixed-size greedy top-$p$ selection).
>
> It is interesting to note that, as a future direction, they mention developing methods to *"further help in incorporating prior knowledge about the reward function"*... which is precisely what vMF-exp sampling is about.

---

### Decision · Program_Chairs · 2025-05-01

**Decision:**

Accept (poster)

**Comment:**

This paper proposes a new method, vMF-exp, for exploration in settings with very large action spaces, such as recommendation systems. The approach uses the von Mises-Fisher distribution to sample exploratory actions based on embeddings, aiming for better computational efficiency than prior methods. A notable strength mentioned by reviewers is its reported deployment and positive results in a real-world music recommendation system. Some also found the paper's explanation of desired exploration properties helpful.

However, reviewers identified several weaknesses. The theoretical explanation relies on an assumption about action embeddings being uniformly distributed, which may not be realistic. Some reviewers found the experimental results unconvincing because the A/B test performance was similar to an existing method (TB-exp), and the paper lacked clear evidence or analysis showing improved exploration. Concerns were also raised that the method might sample isolated or irrelevant actions too often. Issues with writing clarity and the lack of experiments on public datasets were also mentioned.

Overall, the assessment was mixed but leaned cautiously towards acceptance. The method addresses an important practical problem with an interesting approach, even though its justification and testing have limitations.